# Shrinking Lakes, Growing Concerns: Exploring Perceptions of Lake

## **Level Decline as a Prism for Understanding Socionatural Hazards**

- Thomas Vogelpohl<sup>1,2,★</sup>, Desirée Hetzel<sup>1,3,4,★</sup>, Daniel Johnson<sup>5,6,★</sup>, Lena Masch<sup>7,8</sup>, Jesko Hirschfeld<sup>2,5</sup>,
- Thorsten Faas<sup>7</sup>, Peter H. Feindt<sup>1,2</sup>, Jörg Niewöhner<sup>1,3,4</sup>

- <sup>6</sup> Integrative Research Institute on Transformations of Human-Environment Systems (IRI THESys), Humboldt Universität zu
- Berlin, Berlin, 10099, Germany
- <sup>2</sup>Thaer Institute of Agricultural and Horticultural Sciences, Agricultural and Food Policy Group, Humboldt-Universität zu
- Berlin, Berlin, 10099, Germany
- <sup>3</sup>Institute for European Ethnology, Humboldt Universität zu Berlin, Berlin, 12489, Germany
- <sup>4</sup>Department of Science, Technology and Society, Technical University of Munich, 80333, Germany
- <sup>5</sup>Institute for Ecological Economy Research, Berlin, 10785, Germany
- 13 <sup>6</sup>Eberswalde University of Sustainable Development, Eberswalde, 16225, Germany
- <sup>7</sup>Otto Suhr Institute of Political Science, Free University of Berlin, Berlin, 14195, Germany
- 15 <sup>8</sup>Institute for Political Science, University of Münster, Münster, 48151, Germany
- \*These authors contributed equally to this work.
- 17 Correspondence to: Thomas Vogelpohl (thomas.vogelpohl@hu-berlin.de), Desirée Hetzel (desiree.hetzel@tum.de), Daniel
- 18 Johnson (daniel.johnson@hnee.de)

## 19 Abstract

- 20 Groß Glienicker Lake and Sacrower Lake are two lakes in the Berlin-Brandenburg region that are facing significant challenges
- due to declining water levels associated with climate change. In this paper, we report on a study that employed a mixed-method
- approach, incorporating ethnographic research, a household survey and stakeholder workshops, to address: (1) public and
- 23 stakeholder perceptions of these declining lake levels (2) the social structures that interact with these perceptions, (3) the
- willingness to act and perceptions of responsibility, and (4) the local practices for dealing with these challenges. Our analysis
- reveals that lake level loss offers a prism through which such a hazard becomes visible and understandable as, shaped by the
- interdependence of natural and social processes. From this understanding, we develop possible paths forward in governing
- risks adaptively. Such an expanded understanding of lake level loss as a *socionatural* hazard enables the orchestration of more
- comprehensive solutions to such phenomena than is possible solely on the basis of technical remedies to such hazards.

## 1 Introduction

- The capital region of Berlin-Brandenburg in northeastern Germany is known for its numerous surface waters, and the proximity
- of these bodies of water has instilled a sense of familiarity with water in the region's residents (Meyerhoff et al., 2014). In total,

the region counts 33,000 kilometers of watercourses and approximately 10,000 ponds and lakes, which suggests a large supply for daily water needs (Rücker et al., 2019; BUND Brandenburg, 2024). In recent decades, however, these surface waters have come under pressure due to climate change, which has been exacerbated by dry and hot summers such as that of 2018 (Germer et al., 2011; Nützmann and Mey, 2007; Heinrich et al., 2019). One visible parameter of this for the wider public are water bodies affected by falling water levels. The issue of impending water scarcity is now well documented in Berlin-Brandenburg, with both drought events and reduced groundwater recharge mentioned as important factors (Heinrich et al., 2019; Pohle et al., 2025; Francke and Heistermann, 2025). Recent studies at two lakes in the periphery of Berlin, namely Groß Glienicker Lake and Sacrower Lake, have attempted to identify potential causes of the observed lake level decline. These two lakes are the study site for this paper. Investigations of the influence of groundwater trends and subsurface flow (Mahmoodi et al., 2024), water balance models (Somogyvári et al., 2024), and changes in precipitation (Ölmez et al., 2024) suggest that the observed decrease in lake level of Groß Glienicker Lake between 2002 and 2015 can be attributed to lower net precipitation, which, however, cannot fully explain the steep decrease in lake level since 2015. All these studies provide valuable insights into the complexity of managing surface water loss in the region. However, the multitude of factors to be considered also points to the inherent uncertainties and challenges in deriving a single, actionable (climate) adaptation option (Eriksen et al., 2015). For two decades now, matters of sinking water levels have concerned and continue to concern a local population that has grown accustomed to organizing its economic, social and cultural aspects of life along these two surface waters. For local residents at Groß Glienicker and Sacrower Lake, the lakeshores are points of community life, hosting social and cultural activities, educational opportunities, and private and public gatherings. Additionally, day trippers from surrounding urban areas come to swim or hike, making the two lakes spots of recreational value. In this context, lakeside residents refer to initial moments of raising awareness for their lake's issues. On the website of a citizens' initiative, a group of people could be seen on camera standing waist-deep in the water of Groß Glienicker Lake and holding up a big sign that read 'Help', signaling their lake to be in great danger of disappearing (Haid-Loh et al., 2025, p. 4). They had been signaling to local administrations for years that the lake's water level is dropping rapidly. Ever since, Groß Glienicker Lake and Sacrower Lake have become the focus of intense discussions about water management and responsibilities in relation to private and public water use (Degener, 2020). This also opened up discussions about local community life connected to water and highlighted the diverse approaches to the two lakes held by political representatives, visitors and local residents (Kramer, 2021). Debates have also exacerbated existing frictions between local citizens on the one hand, and administrative bodies on the other, over the uncertainties in the search for reasons and solutions to water issues (Görke, 2021; Grote, 2024). In this paper, we address local perceptions and actions from the context of this case study, contributing to the special issue on water issues in Berlin-Brandenburg by addressing the hazard of lake level loss in the region as more than just ecological. In recent decades, climate change impacts such as water scarcity have been increasingly problematized in the literature as not

only an ecological but also a societal challenge, calling into question purely technical approaches to climate adaptation

(Nightingale et al., 2020). In this vein, we will show in the context of Groß Glienicker Lake and Sacrower Lake that measuring

physical indicators such as water levels and property values, while important, does not fully address the challenges seen by people. Rather, it overlooks the complex interplay of factors across physical, political, and cultural domains that contribute to the overall hazard landscape and that need to be considered to comprehensively understand risk in social-ecological systems (see also Spencer and Alexandra, 2024). Therefore, we expand the concept of hazard to include not only the physicality of declining lake levels, but also its perception and the social harms associated with this ecological change. Hazard, defined as any potential source of harm, includes the uncertain origins of these declining water levels and is only one part of the risk analysis equation. Definitions of hazard have evolved over the decades to include a broader, more interdisciplinary understanding of socio-political aspects and how individual and collective perceptions of risk should be recognized for risk governance (Klinke and Renn, 2021). In this paper, the lake levels are a prism through which a broader set of (ecological and social) challenges are diffracted and thus made legible.

Our study is thus positioned within contemporary hazards research, which expands the understanding and management of risk through social science perspectives, by taking into account human behavior, social structures and perceptions, and cultural contexts and practices. Social science perspectives identify how the local community perceive and respond to hazards, which is critical for effective risk communication and mitigation strategies (Cornell and Jackson, 2013). By incorporating these insights, hazard research can develop more comprehensive and inclusive risk management plans that take into account the needs and capacities of diverse populations. This interdisciplinary approach ensures that "hard-path", technical solutions are complemented by "soft-path" strategies that address human and social dimensions, leading to more resilient and adaptive communities (Gleick, 2003; see also Tierney, 2020; Blaikie et al., 2014; Burton, 1993).

Thus, this study adopts an expanded concept of natural hazards as *socionatural* phenomena. A socionatural perspective helps to avoid reifying the nature/society divide and to better grasp and understand the relationality and dynamics of human—nature entanglements in our Berlin-Brandenburg case (West et al., 2020). We examine how the decline of the water levels of Groß Glienicker Lake and Sacrow Lake and the resulting material and social consequences are shaped by the interplay of ecological dynamics, human perceptions, social relations, and institutional frameworks (Boelens et al., 2023) and how people perceive and respond to these challenges in the context of complex social, political, administrative, and scientific structures. This again loops back to how the material changes of the lakes are perceived. While socio-ecological approaches treat society and nature as interacting but separate systems, socionatural perspectives, in contrast, emphasize their inseparability—seeing nature as socially produced and society as materially embedded (Latour 1993).

In the following, we first describe the case study and our mixed-methods approach that draws on a range of social science methods from political science, psychology, economics, and anthropology. The research questions that structure the analytical chapter thereafter concern (1) perceptions of socionatural change, (2) the social structures that interact with these perceptions, (3) willingness to act and perceptions of responsibility, and (4) local practices for dealing with the challenges. The analysis examines the challenges that the growing water publics in Berlin and Potsdam perceive as central to the lake environment and the rationale behind these risk perceptions. This is followed by a presentation of the current strategies of lake residents and their sense of individual and collective agency in (political and environmental) transformation processes. Our analysis reveals

a dynamic and changing approach to the lake as both a public and private space, the need for a governance that supports and enables residents' perceptions of self-efficacy regarding their influence on political decision-making processes, and the problems of adopting one-size-fits-all strategies of action and communication. Finally, we provide insights on how to deal with heterogeneous stakeholder perspectives, drawing on recent studies and lessons learned in the field of adaptive governance.

## 2 Materials and methods

2.1 Case study area in Berlin-Brandenburg Groß Glienicker Lake and Sacrower Lake are two freshwater lakes located in Germany's capital region of Berlin-Brandenburg (Fig. 1). They are situated between the Spandau district of Berlin and the city of Potsdam, the capital of the state of Brandenburg. Both lakes are exclusively fed by groundwater, i.e., they have no surface inflow and thus ultimately depend on the rainfall and the evapotranspiration in the case study area (Somogyvári et al., 2024.). As in the entire Berlin-Brandenburg region, the hydrology of the case study area is fundamentally characterized by dryness. While the annual precipitation remained more or less constant in recent decades, however, the spatio-temporal variability of precipitation patterns and in particular the changes in the frequency and intensity of precipitation extremes have increased significantly in the context of climate change, as historical data show (Bart et al., 2025). Groß Glienicker Lake spans approximately 67 hectares, with a maximum depth of 11 meters (Berlin n.d., Landeshauptstadt Potsdam, 2025a). The lake is divided by the Berlin-Brandenburg border, separating the localities of Groß Glienicke (as part of Potsdam) and Kladow (part of Berlin's Spandau district). The Berlin Wall used to stand directly on the shores of Groß Glienicke because the border between East and West Germany ran through the lake itself from 1945 to 1989. Since the end of the 1990s, the area became more populated and today, both lakes are a popular destination for recreational activities such as swimming, boating, and picnicking (see section 3.3 for more on this). Sacrower Lake is located entirely in Brandenburg next to the Sacrow district of Potsdam. It covers an area of approximately 110 hectares and has a maximum depth of 39 meters. It is surrounded by dense forests that are part of the Sacrower Lake and Königswald Nature Reserve (Landeshauptstadt Potsdam, 2025c). The lake is valued for its biodiversity and tranquil setting, attracting visitors interested in hiking, bird watching, and swimming. Unlike the Groß Glienicker Lake, Sacrower Lake is located in a nature reserve (Flora-Fauna-Habitat) and provides a more natural environment (Landeshauptstadt Potsdam, 2025b).

Figure 1: Map of the case study area. Base map © OpenStreetMap contributors 2025. Distributed under the Open Data Commons Open Database License (ODbL) v1.0. Border layers © GeoBasis-DE/LGB, dl-de/by-2-0 (data not changed). Land cover DE - Sentinel-2 - Germany from the German Aerospace Center (DLR).

In 2024, approximately 20,000 people lived in the immediate vicinity of the two lakes (Landeshauptstadt Potsdam, 2025d; Amt für Statistik Berlin-Brandenburg, 2025), which are both located in the less densely populated area between the German capital the city of Berlin – itself a federal state – and Potsdam, the capital of the federal state of Brandenburg. Groß Glienicke,

a historic village located on the western shore of the lake named after it, was incorporated into Potsdam in 2003, features residential areas, and the community is characterized by a mix of urban and rural settings with increasing residential development in recent years. Located on the eastern shore of Groß Glienicker Lake, the locality of Kladow is a suburban residential area belonging to Berlin. It is connected to Groß Glienicker Lake in the west and the river Havel in the east, as well as other natural and cultural attractions. Sacrow is the smallest district of Potsdam and lies between Sacrower Lake to the west and Havel in the south and east. Formerly, there was a connection to Groß Glienicker Lake to the north and with the Havel. This canal that led to the Havel, which is now obstructed, is referred to as Schiffgraben. Sacrow is surrounded by the forests around Sacrower Lake and the Königswald Nature Reserve, which offer opportunities for outdoor activities and nature conservation.

### 2.2 Mixed-methods approach

Onwuegbuzie, 2004).

In this paper, we follow Bourdieu's work to complement qualitative approaches with quantitative surveys and statistical data to seek a comprehensive understanding of a phenomena (Fries 2009, Lebaron 2009). We seek to "understand" (Bourdieu 1993) socionatural dynamics by combining the qualitative and quantitative study of individual and social perceptions and meaningmaking strategies with the quantitative study of the social conditions that shape those perceptions and are shaped by them. This dialogue between quantitative and qualitative research approaches forms an interpretative approach (Timans et al., 2019). This paper focuses on the decline in the water levels of the two lakes, which was initially declared a hazard by local residents. Labeling it a hazard sparked wider conversations about adaptive measures. In our analysis, we are trying to understand how practices and narratives are shaping the hazard, and how interactions in the social and natural environment again shape perceptions. The mixed-methods approach (Figure 2) in our case study draws on qualitative research (i.e., interviews, participant observation, workshops and focus groups with residents and stakeholders) and quantitative research data (i.e., surveys and survey experiments with the local population and the Berlin-Brandenburg population) (Yin, 2014; Guetterman and Fetters, 2018). Participants in the research included residents of Potsdam (Groß Glienicke and Sacrow) and Berlin (Kladow), stakeholders involved in the research consortium from citizens' initiatives of Groß Glienicke, Sacrow and Kladow, representatives of the Potsdam and Berlin administrations, and visitors from other districts of Berlin. Combining such diverse research methods and sources under the guiding principle of methodological eclecticism helps address complex research

questions by leveraging the strengths of each while mitigating their weaknesses. It emphasizes flexibility and practical

problem-solving over strict methodological purity and is therefore particularly common in the social sciences, where complex

questions often require interdisciplinary and multidimensional analyses (Greene et al., 1989; Kroos, 2012; Johnson and

Figure 2: Conceptual depiction of the mixed-methods approach leading to the overarching themes of analysis through the lens of the socionatural hazard of lake level loss.

#### 2.2.1 Ethnographic methods

As part of ethnographic research, participant observation, qualitative semi-structured interviews, and informal discussions (Shah, 2017) were conducted. In an inductive process to understand current social, political and cultural dynamics, the research questions dealt with local human-water relations in terms of daily activities and discussions at the lakes. Furthermore, this research sought to understand how residents perceive change and act in times of crisis and transformation. This addressed both residents active in citizens' initiatives, concerned but not politically involved residents, and visitors who enjoyed the lakes as recreational areas. Further, it included conversations with local administrators and professional representatives like foresters and fishermen. Discussions at events with representatives from each neighborhood were followed by household interviews and informal interviews with 32 interviewees. This was complemented by participation in action days around the lakes, cultural events, and analysis of online representations and documents. Qualitative data analysis was carried out inductively, with codes grouped into thematic fields (Rädiker, 2023). These were discussed in preparation for the survey and in accompanying the resilience workshops.

#### 2.2.2 Household survey

The insights from the ethnographic research informed the development of a quantitative survey of residents living near Groß Glienicker Lake. Invitations to the survey were sent out using postcards, with 5,000 postcards hand-delivered and 25,000 sent via mailings. Of those invited, 644 residents responded and completed the survey and survey experiment online by scanning a

<sup>&</sup>lt;sup>1</sup> The survey can be made available upon request.

QR-code or following a link to the survey website. While the response rate is low (2.2 %), respondents were heterogeneous in terms of socio-demographic characteristics, perceptions of challenges, and contributions to the lake. A discrete choice experiment assessed trade-offs in lake attributes including water quantity and quality, public paths and facilities and biodiversity against hypothetical costs. Further information on the discrete choice experiment can be found in the Supplement. Retrospectively, the results of the survey aided the generalization of ethnographic research to a wider population.

## 2.2.3 Stakeholder workshops

In addition, a series of four transdisciplinary workshops was held between fall 2023 and summer 2024 with largely the same 8 to 10 representatives per workshop from citizens' initiatives at the lakes and public authorities involved in water management around the lakes as well as from local political organizations and research institutes (see Tab. 1; some individuals represented more than one organization, which is why there are more organizations listed than individuals present at each workshop). This workshop series was built on the resilience assessment framework for farming systems developed by Meuwissen et al. (2019). This participatory and workshop-based approach that allows for the integration of the perspectives of a diverse group of stakeholders, and we adapted it to the needs and specificities of social-hydrological systems, i.e., water-centered humanenvironment systems in which hydrological and social processes influence each other in complex ways (Mao et al., 2017).<sup>2</sup> The first step in this process was to define the study area as such a social-hydrological system based on its main physical and social characteristics and the core and contextual actors that shape it. Against this background, the essential functions this system is supposed to fulfill (see figure 7 in section 3.2.1) were selected and operationalized in a participatory manner by identifying indicators that represent these essential system functions. Subsequently, a longitudinal recollection of the development of the most important indicators has been conducted to identify resilience challenges and discuss the strategies applied in response to these challenges and their current performance in terms of the system's resilience. The third step was to extrapolate the development of the selected indicators into the future in order to identify future resilience challenges. Based on this, the fourth step was to develop strategies for action corresponding to this future trajectory that adequately address both current and anticipated resilience challenges so that the system can maintain or, if it is not currently doing so, be enabled to fulfill its essential functions.

Table 1. Organizations represented throughout the workshop series

| No. | Organization                                                                 | Sector                |
|-----|------------------------------------------------------------------------------|-----------------------|
| 1.  | City Administration of Potsdam (Urban Development, Construction, Economy and | Public administration |
|     | Environment Division)                                                        |                       |
| 2.  | Groß Glienicker Forum                                                        | Local political party |
| 3.  | Local Advisory Council Groß Glienicke                                        | Municipal council     |

<sup>&</sup>lt;sup>2</sup> The structure and the guiding questions of the stakeholder workshop series can be made available upon request.

| 4.  | Pro-Groß-Glienicker-See citizens' initiative                                                  | Civil society                 |
|-----|-----------------------------------------------------------------------------------------------|-------------------------------|
| 5.  | Potsdam Institute of Inland Fisheries                                                         | Science                       |
| 6.  | Sustainability Platform Brandenburg                                                           | Public sustainability network |
| 7.  | Citizens' Advisory Council Sacrow                                                             | Civil society                 |
| 8.  | Freies Ufer                                                                                   | Civil society                 |
| 9.  | Potsdam Institute for Climate Impact Research                                                 | Science                       |
| 10. | Kladower Forum                                                                                | Civil society                 |
| 11. | State Forest Enterprise Brandenburg (Forest enterprise Finkenkrug, Forest district Krampnitz) | Public administration         |

To fully exploit the potential of a mixed-methods approach, we considered the empirical data from all three studies and combined their findings to gain a deeper understanding of the mechanisms behind different local perceptions. We thus used the method of triangulation, although not primarily to cross-validate our empirical findings, but rather to deepen and widen our understanding of the socionatural phenomena at and around the lakes in the sense of Yeasmin and Rahman (2012). Based on this, we generated overarching themes, which we formulated as questions in order to relate results from individual empirical approaches to each other and to the overarching discussions in the joint research project. After several iterations, we arrived at these four thematic blocks that structure the results section below: (1) the socionatural changes that people perceive as challenges, (2) the interrelations of these perceptions with the social, cultural, political, and historical contexts of life at the lakes, (3) the perceived responsibilities and willingness of citizens to act, and (4) the local practices that are already underway in the face of the perceived challenges. This iterative, complementary methodological approach combining qualitative and quantitative methods, allowed us to jointly structure and ultimately answer the research questions regarding the socionatural hazard of lake level loss as a prism through which both the risk of ecological and social challenges become diffracted and more apparent.

#### 3 Results

#### 3.1 What socionatural changes do residents perceive as challenges?

People living around the two lakes emphasize their awareness of the decrease in lake water levels. In initial conversations during ethnographic research on the street, in interviews around kitchen tables and at events, very often the lack of water was the first thing to be emphasized. This then marked not only a starting point for conversations with the researchers, but also the linchpin of the lake residents' own problem analysis. In general, discussions and practices start with this narrative of the danger

of the sinking lake level. However, the challenges associated with this first narrative point to a more complex picture of this hazard. What seemed to be a natural phenomenon, involving the loss of water, prompted people to point towards a wider range of perceived changes at the lake.

The survey of residents living close to the lakes also shows that they are highly aware of diverse changes at Groß Glienicker Lake. Although the problem frames at the lake may vary among residents, groups, and initiatives, the responses in the quantitative survey paint a clearer picture of recognized changes and challenges at the lake. Nearly 80 % of survey respondents indicated that they had noticed a sharp decline in the lake level (Fig. 3). Although 35 % of respondents felt that water quality had declined slightly or greatly, and similar responses were received for bird, insect, and plant diversity, approximately half of the respondents either did not notice a change, or indicated that it had stayed the same. During the joint exploratory phase with interviews to prepare the survey questions, the topic of lake level loss very quickly opened up discussions about further problems of everyday life in this area, putting into context the other challenges mentioned in close relation to the falling lake level (also in the initial interviews with residents). In the survey, strong majorities perceived that the number of visitors (> 80 %), vehicles (> 80 %), and litter (65 %) had increased slightly or greatly. Indeed, many open-ended responses highlighted issues such as littering and visitor pressure resulting from lockdown measures during the Covid pandemic, which also led to trampling of shoreline vegetation, unsanitary infrastructure and wild bathing.

Figure 3. Perceived awareness of changes at Groß Glienicker Lake from the survey (N = 644).

This general awareness of current changes translates into concerns about the future for the Groß Glienicker lake, in which almost 90 % of the survey respondents reported the water level to be a large or very large future challenge for the lake (Fig. 4). Furthermore, people living in this area equally consider higher temperatures, less rainfall, and climate change as challenges. More than half believed private water use to be a high or very high danger for the lake, and 45 % considered groundwater extraction by water utilities to be an issue. In addition to water quantity, the residents identified water quality as a challenge for the future (60 % high or very high). Further responses from the interviews revealed that the residents had an understanding of ecology and the impact of human activity. They recognized the importance of maintaining good water quality through

proper water levels and natural vegetation along the shore. Here, residents see pressures from the population and rising number of visitors and highlighted the ecological challenges with people not taking care of their surroundings (i.e., trampling of reed vegetation along shore by the increasing number of visitors). The ecosystem was one of the main concerns expressed in conversations. However, these are weighed against the utility value that residents in this region derive from them. This leads them to develop an appreciation for their surroundings, which they then prioritize preserving. We will go into further detail about this in section 3.4.

Figure 4. Perceived future challenges or dangers for Groß Glienicker Lake from the survey (N = 644).

Interestingly, the perceived lake-related risks expressed in the survey that are associated with climate-related changes may not be consistent with the overall perception of climate change risks in the region. The majority of respondents (over 80%) perceived climate change to be a significant future challenge for the lake (see Fig. 4). However, a small proportion of respondents did not consider climate change to be an issue, or were unaware of it. In another set of questions not directly related to the lake, respondents indicated their perceptions on a scale from very true to very false concerning the statements "Climate change is mainly caused by humans," "There are many different scientific opinions about climate change," and "The media exaggerate the possible effects of climate change" (Fig. 5). These responses were also used in the choice model to examine the direction of the effect such perceptions have on the willingness to pay for changes in the attributes of the choice experiment (Section 3.3). The majority of respondents (almost 80 %) in the household survey believe that climate change is primarily caused by humans, and almost 50 % agree that this statement is very accurate. Moreover, as can be seen in Figure 5, one-third or less believe that the media exaggerate the effects of climate change and that the scientific community is divided on climate change.

Figure 5. Climate change skepticism among residents from the survey (N = 644).

This suggests that the sources of risk perception, such as the role of climate change in creating or causing the challenges at the lake, may not always be potential influencing factors. While most people recognize the connection between climate change and the challenges facing the lake, some remain skeptical about anthropogenic climate change and may attribute the changes to other causes. This could impact people's willingness to change their behaviors, either individually or collectively. When talking to people, topics connected to global warming, water scarcity and changes in weather patterns are connected to current climate politics. While some refrain from being 'distracted' by this topic, it still brings up matters of anthropogenic influences of the environment for them. Human behavior that does not take care of the surroundings was strongly criticized by residents. People would rather rely on what they experience in their everyday lives. In this way, life at the lake becomes a first-hand experience of discourses that are otherwise mentioned in political debates or in the media. They turn to other examples of affected water bodies in the region and search for solutions to preserve the waterscape.

This general awareness of ongoing socionatural changes at the lakes - albeit with sometimes different explanations for them - established in the ethnographic interviews and the quantitative survey is very much in line with the assessment of future challenges to the lakes as a social-hydrological system during the participatory resilience workshops. While the resilience workshops differed to the other research parts in terms of the aim and approach, the analysis leads to a complementary depiction of the challenges. The set of challenges in the quantitative survey was predefined in accordance with the explorative interviews in collaboration with the ethnographic research. Similarly, the challenges rated in the stakeholder workshops were pre-selected based on desk research and joint interviews with political and ethnographic focus and then verified by the participants during the first workshop. Afterwards, participating stakeholders rated the challenges identified this way on a scale from getting smaller (-2) to staying the same (0) to getting bigger (2), as shown in Fig. 6. Although there is some heterogeneity among the responses, the general result of this assessment is that all these aspects, which were already perceived as challenges, were expected to become even more problematic (with the exception of population growth).<sup>3</sup>

<sup>&</sup>lt;sup>3</sup> In addition, there was confusion among workshop participants during the assessment as to whether population increase included only the communities in immediate vicinity or also the wider surroundings of the lakes. During the discussion after the assessment, however, it was clarified that participants expect considerable population growth in the nearby northern areas of Potsdam (see also section 3.2), which is actually expected to aggravate the status of the socio-hydrological system of the

Figure 6. Average stakeholder perception of future challenges becoming smaller (-2), staying the same (0) or becoming larger (2) (N = 8).

User behavior

These results of the stakeholder workshops concerning the biggest future challenges for the socio-hydrological system align with the residents' perceptions of the biggest changes and challenges around the lakes collected in the survey, in that they point to a similar set of ecological changes (such as higher temperatures, more dryness/evaporation, less rain, and altered precipitation patterns) and social developments (such as increased number of visitors, population growth, and harmful user behavior). In addition, the stakeholders' assessment of the biggest future challenges reveals a perceived governance failure around the lakes. They both identify a lack of resources in administration, (inadequate/unclear) management structures/responsibilities, and a lack of or contradictory regulations as among the currently largest challenges. They also expect these three governance-related challenges to become even more relevant in the future (especially the lack of resources in administration).

## 3.2 What social structures underlie and interact with these perceptions?

The previous chapter outlined the perceptions of the changes and challenges associated with the two lakes and provides a clear picture that, overall, residents and stakeholders strongly perceive changes and recognize future challenges for the lakes. This strong perception is due to the central role that both lakes play in social, political and cultural life around them, as field visits and conversations revealed. Both lakes are open to the public in some areas and are therefore highly valued. However, access is restricted in other areas due to private ownership or environmental protection zones. Sacrower Lake is used for local recreation, and walking and running are popular activities there. Swimming is tolerated in a designated bathing area due to the

lakes. Taking this into account, it can be concluded that all of the currently identified challenges are actually expected to become more severe in the future by the participating stakeholders.

landscape protection regulations. Groß Glienicker Lake has become even more involved in community activities for local residents, and its designated bathing areas make it a popular meeting place. Environmental education, political education programs on German history, festivities and gastronomy are always related to the lake. Here, too, social family experiences are linked to the history of the lake. In everyday life, the lake is a meeting place for walkers, runners, swimmers and local anglers who are members of the fishing club. The above-mentioned perceptions of socionatural changes as (future) threats thus take place in a wider social, cultural, political or historical context.

We now turn to those contexts that are important to understand in order to see how certain perceptions have been framed. This gives insights into the general preferences of residents and visitors regarding life at the lakes, which we found to be related to the social and demographic structure of the lakeside settlements, the political history of the area, and the fragmented administrative responsibilities around the lakes.

#### 3.2.1 General preferences regarding the lakes and their future

When stakeholders were asked about the essential functions of the lake as a social-hydrological system during the workshops (see section 2.2), an attractive and healthy living space offered by the lakes and the social participation they enable were rated highest (see Fig. 7).<sup>4</sup> The residential area of Sacrow, Groß Glienicke and Kladow is currently characterized by its proximity to the city, but its distance from the urban center gives it a village feel. The necessary shopping facilities are complemented by only a few cafés and a handful of restaurants. A restaurant on the northern shore of Lake Sacrow has been converted into a temporary event venue, and the café on Groß Glienicke Lake is usually open on weekends during the summer months. Both residents and visitors therefore emphasize the tranquility of the area and the natural surroundings by the water. The preference to keep this healthy environment intact is thus related to the rising number of people and visitors and their (allegedly) harmful behavior. Closely related to the attractiveness of living space and social participation is the environmental health of the system, represented by the functions of biodiversity and climate protection as well as the conservation of natural resources. These functions are perceived as being indirectly challenged by the increasing number of users and their harmful behavior and - more importantly - directly challenged by the decline in lake water levels and quality. Lastly, the meaning of the lakes with regard to the preservation of the historical and cultural identity of the region plays a meaningful role in the participants' evaluation of the essential functions of the social-hydrological system.<sup>5</sup>

<sup>&</sup>lt;sup>4</sup> Although they differ in concept and survey method, we consider the preferences for future changes at the lakes collected in the quantitative survey and the essential functions of the social-hydrological system collected in the stakeholder workshops to be largely identical in meaning. Both are concerned with what is essential to the participants with regard to the lakes now and in the future.

<sup>&</sup>lt;sup>5</sup> Although ranked lower on the list of essential functions shown below, the lake also has an economic relevance for lake-side property owners. Property values around the lakes have risen considerably in recent decades due to the influx of population into the area (see below). This value depends at least partly on the status and existence of the lakes, so the loss of lake levels is also an economic threat to some of the residents, especially to those whose property is still directly on the lakes.

Figure 7: Ranking of essential functions of the social-hydrological system of the lakes from the stakeholder workshops (N = 8).

These essential functions of this socio-hydrological system align with the preferences for future changes at the lake found in the quantitative survey (Fig. 8). Nearly 100 % of respondents indicated the importance of water level stabilization as important or very important, and almost 80% of respondents perceived future improvements in water quality to be important or very important. Improved waste disposal was similarly ranked by over 90 % of respondents. The subsequent issues with the recognized increase in the number of visitors above were also reflected in the importance or high importance of better enforcement of rules and more provision of information about the lake as a natural environment. As a result, residents do not assign a higher importance to more cafes, shops, restaurants or parking spots. Despite this, residents do see value in access to the lake and transportation in general, with just under 50 % of respondents assigning importance or high importance to better public transportation and bike paths.

Figure 8. Preferences for future changes at Groß Glienicker Lake from the survey (N = 644).

Through this, we can distill two overarching themes: first, participants highly value the ecological status of the lakes and the adjacent water-dependent ecosystems. Second, they care about the social services these ecosystems provide, such as an attractive living space and social participation. However, the economic value of the lakes is less important as reflected in the fact that investments in public infrastructure are only valued if they are perceived to improve the ecological status of the ecosystem and not its tourism value. Rather, they are critical of further changes that enable greater public access, which is consistent with the results presented in the previous section where this aspect was identified as one the main challenges to the lakes and life at the lakes.

#### 3.2.2 Social and demographic structure

The preferences of residents regarding the lakes are partly shaped by demographic developments and the social structure of the region, which therefore also influence the perceptions of socionatural changes around the lakes. With regard to the quantitative survey, the sample from the study area was older (55 years on average) than the average population age of Berlin (42.8 years; Amt für Statistik Berlin-Brandenburg, 2024a) and Potsdam (43.2 years; Amt für Statistik Berlin-Brandenburg, 2024b). Overall, 66 % of the sample had at least a first degree and the average net income per month was over € 5,000, while the average monthly gross incomes of fully employed residents are approximately € 4,500 for Berlin and € 3,600 for Brandenburg (Amt für Statistik Berlin-Brandenburg, 2024c). Almost 60 % of the sample were employed at least part-time during the survey. The average household size was about 2.7 persons, higher than in Berlin (1.9 persons, Amt für Statistik Berlin-Brandenburg, 2024d) and Potsdam (1.9 persons; Amt für Statistik Berlin-Brandenburg, 2024e). Single-family dwellings form the main component of the settlement structure, especially near the lakes. These figures are in line with the official data on the demographic and social structure in the region, which on both sides of the lakes is characterized by a relatively high

proportion of older, wealthier people with a relatively high social and educational status with stable dynamics (Bezirksamt Spandau von Berlin, 2021; Landeshauptstadt Potsdam, 2023).

This demographic structure of the case study area has not always been like this. In fact, it changed significantly after the German reunification in 1990, especially on the Potsdam side. The population of Groß Glienicke, for example, more than tripled from about 1,500 inhabitants in 1990 to about 5,000 inhabitants today, mainly because it became an attractive area for relatively affluent elderly people and families from Berlin, but also from other parts of former West Germany. Accordingly, more than 70 % of the (mostly detached and semi-detached) houses in Groß Glienicke were built after 1990 (Landeshauptstadt Potsdam, 2023: 66-67). The population in Groß Glienicke and Sacrow is expected to keep growing steadily in the future, although at a slower pace than before. However, in 2019, the city of Potsdam decided to develop a new city district on a former military site close by that is expected to be home to 10,000 inhabitants until 2040. Kladow is also characterized by a relatively affluent population that has grown significantly in the last decades, especially in the vicinity of the lake. However, this growth is expected to come close to a halt in the near future (Bezirksamt Spandau von Berlin, 2021).

Thus, even though the population directly at the lakes will probably not grow significantly anymore in the future, the overall usage pressure on the lakes from people living directly at or near the lakes and using them for leisure and recreation will probably continue to grow. This will affect both the ecological status of the lakes and adjacent ecosystems and how people in the area perceive and attribute corresponding changes. These socio-demographic developments and the growing number of visitors at the lakes (see below) have raised concerns among residents about the additional pressure these developments will put on the lake sites, as well as their ability to cope with this pressure given the lack of appropriate infrastructure. Although people support making the lakes accessible to everyone, they fear that existing paths will be ignored and that littering will increase.

In addition, both lakes have become a popular tourist destination in the last decade. During ethnographic research, local residents pointed out that people were particularly attracted to the nearby lakes during lockdowns due to the coronavirus pandemic in 2020 and 2021. During this period, local recreation areas became increasingly popular and people from the city traveled to nearby lakes. This increase in visitors is associated with several challenges and associated preferences for future changes, such as the growing challenge of bad user behavior (see section 3.1, Fig. 6) or the desired marking and inspection of designated bathing areas, sanitary and waste disposal facilities, and traffic management (see section 3.1, Fig. 8). Although visitor numbers peaked during the pandemic, the growing population of the Berlin metropolitan area raises concerns that the pressure of use will continue to increase (see section 3.1). While there is an ongoing debate, also among residents, whether public access to the lakes should be partly restricted or expanded, there is a consensus that use of the lake must be accompanied by visitor guidance and public infrastructure such as toilets and waste disposal. We will address these matters of administrative responsibilities further below.

#### 3.2.3 Political history of the region

The current demographic and social structure of the region, which considerably influences local perceptions of socionatural changes and challenges at the two lakes (as shown above), is the result of development over the last century. From a rural area with a settlement history dating back to the 13th century, Groß Glienicke developed into a village inhabited by middle-class city dwellers at the beginning of the 1920s. It was then shaped by separation and finally German reunification (Lehmberg and Toreck, 2007). These historical milestones have been raised both in interviews as well as in workshop group discussions. In addition, they characterize not only the perception of current sociocultural changes around the two lakes, but also the current political and administrative structure. This structure is relatively fragmented with regard to responsibility for water management in the lakes' catchment area and who is responsible for addressing these changes (see sections 3.2.4 and 3.3 below).

The division of Germany (1945-1989) resulted in the border running directly through Groß Glienicker Lake, with the Berlin Wall running along the shore of the lake in Groß Glienicke. One interview partner underlined the importance of the Groß Glienicker Lake as a historical monument, because it brings to mind all the political quarrels and connects its residents also to Sacrower Lake and the politics of the whole region. As the ethnographic research revealed, this had a significant impact on the daily lives of those living around the lake, particularly those on the western side. The wall cut off Groß Glienicke residents from access to the lake, forcing them to go to Sacrower Lake instead. The local fishing club of Groß Glienicke is therefore still located at Sacrower Lake. Demarcation and separation characterized village life (Diedrich et al., 2022), given that people in Groß Glienicke were living in a zone only possible to enter with permits (fieldwork, 2023).

After the fall of the Berlin Wall in November 1989, Groß Glienicke, which was still an independent municipality at the time, regained public access to the lake, and the former border patrol path on this side of the lake was supposed to become a continuous lakeside path. However, due to legal loopholes, there is an ongoing legal conflict with private owners over public access to this path (Zschieck, 2021). Sacrow has been part of the city of Potsdam since the 1930s, while Groß Glienicke was only incorporated in 2003, becoming one of Potsdam's 32 districts (Franzke, 2021). Kladow, located on the eastern side of Groß Glienicker Lake, was incorporated into Berlin in 1920. Starting in 1929, the eastern shore of the lake was planned as a single-family housing estate for Berlin residents. In addition to their detached house in the estate, the new property owners were also able to purchase private access to the lake with a jetty, which attracted many Berliners to Kladow to settle here as this enabled middle-class Berlin families to construct a weekend residence on the lakefront (Schmiedecke, 2002, Bankmann, 2015). During the division of Germany, the residents of Kladow still had access to the lake, although it was - with the exception of two public bathing areas - predominantly private, which is still the case today. Thus, the ethnographic research confirmed that the German reunification did not change nearly as much on the eastern, Berlin/Kladow side of the lake as it did on the western, Potsdam/Groß Glienicke side (fieldwork, 2022).

To this day, this history is reflected in family stories and biographical narratives. Conversations about the lakes during the ethnographic research always included references to the lakes as a political monument. This history of the area interwoven

with wider political heritage is still part of cultural and educational programs and events along the lakes. Thus, it provides an important interpretive context for the perception of socionatural changes and challenges of and around the lakes (see section 3.1). It also directly impacts the administrative structures surrounding the lakes and the associated responsibilities, which we will turn to in the following section.

German reunification has meant that the Groß Glienicker Lake is no longer divided by a national border. However, it is still

#### 3.2.4 Administrative Fragmentation

divided by the border between Berlin and Brandenburg - i.e., the border between two German federal states - which, to this day, results in significantly fragmented political and administrative governance structures. This is because the responsibility for water management in the lakes' watershed is distributed among several Berlin, Potsdam, and Brandenburg authorities. In addition, the administrative structures on both sides of the lakes are themselves already considerably fragmented, as was mentioned numerous times during the stakeholder workshops (see also Grote 2024). In the context of the EU Water Framework Directive (WFD), one of the states takes the lead responsibility for cross-state water bodies. In this context, Berlin is responsible for all lakes where the state border runs through the middle, i.e., also the Groß Glienicker Lake (SenUMVK, 2021, 13). Responsibility for water management in Berlin is shared between the central state administration and the districts. The Senate Department for Urban Mobility, Transport, Climate Action and the Environment is responsible for groundwater and for the maintenance of all water courses and first-order standing water bodies (ibid.). The district authorities, however, are responsible for the conservation and maintenance of second-order standing water bodies such as the Groß Glienicker Lake. This task is carried out by the district offices (Bezirksämter) responsible for nature conservation and green spaces (SenMVKU, 2025). Thus, the Bezirksamt Spandau is responsible for the Groß Glienicker Lake, while the Senate is responsible for the groundwater in the catchment of the lake and thus also the aquifers traversing the lakes, including its parts belonging to Brandenburg. Since Sacrower Lake belongs to Brandenburg in its entirety, only Brandenburg authorities are responsible in terms of the surface water. In general, surface waters fall into the responsibility of the Ministry of Agriculture, Food Industry, Environment and Consumer Protection, which is the highest water authority, and the State Office for the Environment, which is the upper water authority (MLEUV, 2020). The lower water authority in this case is part of the city administration of Potsdam, which is responsible for monitoring and protecting water bodies from impairment according to the federal and state water law (MLEUV,

<sup>6</sup> Standing water bodies in Berlin are administratively divided into first-order standing water bodies (navigable) and second-order standing water bodies (non-navigable).

2025). The Schiffgraben, a water stream connecting Sacrower Lake with the nearby river Havel, falls within the area of

responsibility of the water and soil association Nauen, which mows the embankments by hand once a year and clears the ditch

of dead wood and debris.<sup>7</sup> The bridge across the Schiffgraben, however, belongs to the city of Potsdam (MLUK, 2020, 30).

<sup>&</sup>lt;sup>7</sup> In Germany, water and soil associations are self-governing corporations under public law that perform water and soil management tasks in the public interest and for the benefit of their members (mainly public and private landowners). They

These fragmented governance structures, in the eyes of the residents and stakeholders, hinders the administration as a whole from taking appropriate actions to solve lake-related problems such as the (perceived) residents'/visitors' behavior and lake level loss or to mitigate their effects. Based on their (perceived) past experience, administration is not to be trusted to solve the lake-related problems, which is why the lack of regulations and their enforcement as well as inadequate management structures were perceived to be salient future challenges both in the quantitative survey and during the stakeholder workshops (see section 3.1).

The social structures depicted in this section revealed that perceptions of changes and challenges are shaped by the general preferences of residents and stakeholders regarding what is considered essential and desirable. Moreover, the perceptions are linked to demographic characteristics and the social structure of the communities around the two lakes through the political history of the region as well as the associated, fragmented administrative structure. In light of the emerging inability to act, the question arises as to how responsibilities are perceived and whether there is a willingness to actively participate in addressing the challenges.

## 3.3 What are people willing to do and who do they consider responsible?

Observing the rapid lake level loss of both lakes for a decade, people started discussions about what this loss would mean to them. While economic conditions such as house price developments also shape individual preferences, there are also strong indicators for common approaches of caring for the lakes. Results of the quantitative survey suggested that residents of the study region (8 km radius) are in fact willing to take some things into their own hands (Fig. 9). Approximately 95 % of respondents indicated that they observe nature conservation rules and take care of their own trash when spending time at the lake. Furthermore, over 90 % of respondents reported not entering natural shore areas, one of the critical points brought up by the residents about the damaging effects of the visitors. Interestingly, almost 60 % of respondents agreed to reduce their personal water consumption, even though the perceived change in the lake level was high and stabilizing the level was perceived as very important. In terms of contribution types, respondents appear more willing to take action regarding their own behavior and consequences (e.g., obeying rules, collecting their trash, and using provided toilets). However, they are less willing to contribute to compensating for the actions of others (e.g., voluntary environmental actions, collecting trash, and informing others about nature conservation rules) or hazards, such as water level decline, that are more likely caused by climate change impacts.

have their legal basis in the Federal Water Association Act (WVG) and the corresponding implementation laws of the federal states.

Figure 9. Indicated willingness to contribute among the respondents to the improvement of the situation at Groß Glienicker Lake from the survey (N = 644).

The willingness to contribute was also reflected in the results of the discrete choice experiment. A conditional logit model was used to model the sample's preferences for changes to the lake in terms of "maintaining or improving water levels", "improving or allowing water quality to deteriorate", "making the lakeshore path public or private", "adding garbage cans or garbage cans and clean toilets", and "improving or reducing biodiversity". Results of the interaction model (Tab. 2) indicate a high willingness to act in terms of making a monetary contribution to the improvement of several attributes. Moreover, the negative willingness to pay (WTP) indicates unfavorable attributes, and the magnitude shows the compensation that individuals would require to accept such a decline in the state (i.e., 70.29 € per household would need to be compensated for the removal of the currently partially public lakeside path). The responses concerning climate change skepticism (Section 3.1) were included in the model as interactions with the attributes after re-coding the first two items, the average of the responses across the three items was calculated before interaction with the attributes. Significant interactions with the climate change skepticism responses were found and demonstrated that on average, higher climate change skepticism leads to a lower WTP for maintaining or improving water levels, improving or allowing for a decrease in water quality, fully opening the lakeside path to the public and improvements in biodiversity.

510

Table 2. Willingness to pay (WTP) estimated through the conditional logit model including interaction terms from the survey.

| (Interaction) Variables of the model                                                                                                                                                                                                                          | WTP per household and year                                                    | Standard error         |
|---------------------------------------------------------------------------------------------------------------------------------------------------------------------------------------------------------------------------------------------------------------|-------------------------------------------------------------------------------|------------------------|
| Alternative specific constant                                                                                                                                                                                                                                 | -62.27***                                                                     | 18.57                  |
| Higher water levels                                                                                                                                                                                                                                           | 422.10***                                                                     | 32.849                 |
| Maintaining water levels                                                                                                                                                                                                                                      | 303.55***                                                                     | 30.851                 |
| 4 m visible depth                                                                                                                                                                                                                                             | 67.13***                                                                      | 21.979                 |
| 1 m visible depth                                                                                                                                                                                                                                             | -3.08                                                                         | 23.23                  |
| No public lakeside path                                                                                                                                                                                                                                       | -70.29***                                                                     | 10.947                 |
| All public lakeside path                                                                                                                                                                                                                                      | 88.77***                                                                      | 19.822                 |
| Trash cans and toilets                                                                                                                                                                                                                                        | 45.18***                                                                      | 11.12                  |
| Trash cans                                                                                                                                                                                                                                                    | 19.47*                                                                        | 10.688                 |
| Biodiversity deterioration                                                                                                                                                                                                                                    | -116.73***                                                                    | 11.307                 |
| Biodiversity improvement                                                                                                                                                                                                                                      | 141.60***                                                                     | 20.976                 |
| Higher water levels * climate change skepsis                                                                                                                                                                                                                  | -46.10***                                                                     | 7.22                   |
| Maintaining water levels * climate change skepsis                                                                                                                                                                                                             | -31.08***                                                                     | 7.643                  |
| 4 m visible depth * climate change skepsis                                                                                                                                                                                                                    | -21.39***                                                                     | 6.341                  |
| 1 m visible depth * climate change skepsis                                                                                                                                                                                                                    | -14.08**                                                                      | 6.941                  |
| All public lakeside path * climate change skepsis                                                                                                                                                                                                             | -23.75***                                                                     | 5.72                   |
| Biodiversity improvement * climate change skepsis                                                                                                                                                                                                             | -32.33***                                                                     | 5.926                  |
| Model characteristics                                                                                                                                                                                                                                         |                                                                               |                        |
| Log likelihood                                                                                                                                                                                                                                                | -4801.68                                                                      |                        |
| Null log-likelihood                                                                                                                                                                                                                                           | -5476.62                                                                      |                        |
| Pseudo R2                                                                                                                                                                                                                                                     | 0.12                                                                          |                        |
| number of choices                                                                                                                                                                                                                                             | 5136                                                                          |                        |
| number of respondents                                                                                                                                                                                                                                         | 642                                                                           |                        |
| AIC                                                                                                                                                                                                                                                           | 9639.36                                                                       |                        |
| BIC                                                                                                                                                                                                                                                           | 9757.15                                                                       |                        |
| 1 m visible depth * climate change skepsis All public lakeside path * climate change skepsis Biodiversity improvement * climate change skepsis Model characteristics Log likelihood Null log-likelihood Pseudo R2 number of choices number of respondents AIC | -14.08** -23.75*** -32.33***  -4801.68 -5476.62 0.12 5136 642 9639.36 9757.15 | 6.941<br>5.72<br>5.926 |

Significance levels: \*\*\*p 

Figure 10. Perception of responsibility by the residents concerning who should implement improvements at the Groß Glienicker Lake from the survey (N = 644).

While residents showed a general willingness to act, either through financial contributions or behavioral changes, the interplay of perceived challenges and perceived responsibilities among various stakeholders emphasizes the need for collaborative and multi-level governance to address the social and ecological hazards that are diffracted through the prism of the perception of lake level decline.

#### 3.4 What are local practices of dealing with perceived challenges?

Loh et al., 2025).

In the previous section we underlined the importance of understanding how diverse actor groups respond to pressing social and ecological challenges. In general, local problem analysis has created a new public sphere that divides into two subgroups. One subgroup consists of people concerned about current developments in their environment who do not feel empowered to take action. The other subgroup consists of active community members who research causes and possible solutions discussed in political arenas and build political pressure by asking questions. In this context, while some ask themselves what their contribution could be, others got active and organized themselves in several initiatives. For them the hazard formed new connections of people through and over the waters. How do the different actor groups, especially the latter, practically deal with the perceived challenges and responsibilities?

In what follows, we refer to the citizen initiatives with which we collaborated in workshops and met during site visits, as well

as to their practices for dealing with the challenges associated with declining water levels. These initiatives also influenced

how residents perceived the topic, as they became spokespersons, set up websites and contributed to a joint publication (Haid-

## 3.4.1 Initiatives on site

Despite their relatively small size and population, the three residential areas around the lakes have a flurry of citizens' initiatives concerned with the lakes and their ecosystems. The representatives of these initiatives base their approach on their own research and mitigation activities at the lakes. In their view, this also helps them retain agency and compensate for a lack of structural support. For example, as one of the smallest parts of Potsdam, the remote Sacrow has a very active and influential citizenry for all matters relating to this neighborhood, and stakeholders underlined the importance of taking care of their lakes. Their own research has made the people experts in their own terms concerning water management. This resulted in a newly founded 'water working group', a cooperation across all citizens' initiatives from all sides of the lakes, to gather and share information about scientific and political approaches. Popular topics included the transfer of water from rivers to lakes for storage to provide the metropolitan region with freshwater. Both options were based on comparable lakes and approaches to water issues in the region. To achieve immediate results, the focus was placed on water use and extraction by residents and the nearby waterworks. Although private consumption is difficult to track, the stakeholders were surprised that figures from the waterworks were not available (although numbers show that these have a minor effect on water levels). Above all, there is growing concern among this water working group that the lake continues to be regarded as an infinite water resource and neither local people and visitors nor institutions respect its limitation and thus rethink their water use, including groundwater use.

In addition to these discussions, people are turning to on-site, concrete activities. One example is the lakeside working days: One such activity at Sacrower Lake brought together local residents and a few representatives from the Potsdam administration. Due to the increasing number of visitors to Sacrower Lake, which is not a designated bathing lake, more and more guests are entering the protected shores of the Flora and Fauna Habitat. The local forester is concerned about the ecological balance of the lake and has set himself the task of surrounding the lake with wooden fences. He explained that the goal is not to prevent potential swimmers from coming to the lake; therefore, he leaves gaps in the wooden fence. Rather they wanted to create awareness that "you can't just go straight to the lake" (stakeholder representative). While the barriers were being erected on the shore, small groups of walkers passed by and suddenly complained about the new boundary, interpreting it as a sign that the lake dwellers wanted the shore to themselves. Aware of this ambivalence and their privileges, the lake dwellers argued that they had also taken on the task of caring for their lakes. When asked, one of the younger volunteers on the day explained: "I'm here today because we have to do something so that this lake doesn't disappear. We like living here". However, some of them also indicated that their privilege of living close to these lakes and being able to swim in them comes with certain responsibilities on their own property and on city property (fieldwork, 2022).

On the Kladow side of the Groß Glienicker Lake, the chairwoman of the local citizens' initiative to protect the lake also believes that private ownership goes hand in hand with protecting the natural environment by maintaining the shoreline. There has been a growing conflict over whether private jetties should be removed. This local discussion used to focus solely on one's access to the lake and its ecological effects, but it has now expanded to include preventing further siltation and weed growth along

the shoreline. There is now a proposal that private owners should also be responsible for monitoring water levels, plant species and animal populations. In their view, this is a task that the district office is not fulfilling. Here, too, some residents see themselves as responsible. However, there is no consensus on what this kind of landscape management should look like, and there is no governing authority.

All in all, the activities on the ground relate to the formation of new groups for the exchange of information and joint strategy development. Here, the lakes become a connecting element. Nevertheless, there are differences in the degree of commitment. While on the one hand, those involved in groups are annoyed about too little participation, others are overwhelmed by the issue and wonder how their own actions can play a role here.

These activities and discussions among active citizens around the lakes have intrinsic and political value. They address local

administrations and politicians, aiming to save the lakes as ecosystems and historical and cultural heritage sites. For example,

the citizens' initiative at the Kladow side of Groß Glienicker Lake mentioned above has issued a 16-page petition to the district

#### 3.4.2 Local citizens and administration between confrontation and dialogue

problems perceived by the citizens' initiative.

office in Spandau in 2019, in which it documents the environmental degradation and pressures on the natural ecosystems on the side of the lake caused by the drying up of large areas of shoreline and the increasing usage pressure from visitors. Based on this, they urged the district office to withdraw the request to dismantle the private jetties in the lake, as these have a positive effect on the riparian ecosystem. Instead, they urged the district office to work more closely and trustingly with the lake residents to improve the lake's ecology together. In this context, they also offered to become active and take on certain tasks themselves, such as regular observations and measurements on site (e.g., photographic documentation) as well as active maintenance measures under the guidance of the specialist authority (BiPGGS, 2019). In 2023, they documented the changes in the water level and the associated degradation of flora and fauna around the lake over the past decade as part of an exhibition entitled "Our lake, (not) a climate victim?", providing information on both the scientific basis and the suspected causes, as well as making demands on politicians and the municipal administration (BiPGGS, 2025). Another example is the Citizens' Advisory Council for Sacrow, a body initiated by the citizens of the village themselves to represent their interests vis-à-vis the city administration of Potsdam. Among other things, this group has focused its work on the Schiffgraben, a connecting ditch between the Sacrower Lake and the river Havel. This Schiffgraben has been interrupted since the 1980s by a meanwhile dilapidated retaining structure, which causes the Schiffgraben to become increasingly silted up and smelly, thereby also increasingly jeopardizing the water quality of Sacrower Lake. In this context, the group has been trying for years to understand and fix this problem, both independently and in cooperation with the local and state administration. For example, it has independently commissioned a nature conservation report to discuss whether and how the dam on the Schiffgraben should be renewed to retain water in Sacrower Lake, prevent eutrophic Havel water from entering the lake and ensure fish passability. The citizens' initiative repeatedly approached the responsible authorities at local and state level with specific local expertise such as this in order to enter discussions with them and collaborate on solutions to the

With actions such as these, on the one hand, the citizens' initiatives encounter rejection and inaction on the part of the authorities, at least according to their own perception, which leads to considerable frustration and resignation. The different points of contact resulting from the overlapping responsibilities (see section 3.2) continue to create uncertainty among committed citizens. As we described earlier, the responsibilities for surface water and groundwater are distributed differently and are also dependent on long-term cooperation. As a result, committed groups encounter ambiguous areas of responsibility, which in turn pass these commitments on to other bodies. This results in what local residents like to call an 'administrative Mikado', in which responsibilities overlap and therefore the questions and demands of the committed public end in a dead end. In discussions with the administration, this is mainly due to the range of different tasks as well as to staff shortages (see also section 3.1). On the other hand, the actions of the citizens' initiatives also lead to more and better understanding and cooperation, both among themselves and between citizens' initiatives and the local authorities. As shown above, the overarchingly problematic situation of the lakes has led to the citizens' initiatives from the three residential areas networking more closely and also planning and carrying out joint campaigns. In this sense, the lakes and their problems have served as a kind of boundary object for new societal cooperation, not only among the citizens' initiatives, but also between citizens' initiatives and the municipal administration (Franco-Torres et al., 2020). In reaction to the manifold problems around the lakes and the discontent of the citizens, the district office of Spandau and the city administration of Potsdam teamed up in 2021 to start a citizen dialogue on the two lakes that aimed at jointly preparing the call for tender for a feasibility study on future options for water management of the two lakes. The goal of this dialogue was to work out the requirements for such a feasibility study together with the citizens involved to represent all their viewpoints

In reaction to the manifold problems around the lakes and the discontent of the citizens, the district office of Spandau and the city administration of Potsdam teamed up in 2021 to start a citizen dialogue on the two lakes that aimed at jointly preparing the call for tender for a feasibility study on future options for water management of the two lakes. The goal of this dialogue was to work out the requirements for such a feasibility study together with the citizens involved to represent all their viewpoints and preferences in the process and in the final feasibility study. The dialog process consisted of four meetings between spring 2022 and summer 2023, in which the relevant people, groups, institutions and organizations discussed and agreed on which aspects of a feasibility study need to be examined, presented and demonstrated in order to develop a clear target perspective for the future development and future handling of the two lakes. The fundamental aim was to promote and facilitate the comprehensibility of the results and the broadest possible agreement of all stakeholders on the objectives of a feasibility study. The call for tenders for the feasibility study was published in August 2025. The study will be conducted in 2025/2026 and will serve as the basis for implementing concrete measures beginning in 2027 (Landeshauptstadt Potsdam, 2025e). While it is not yet clear how this process will end and what fruits it will ultimately bear (which is why it is still viewed with much skepticism), it carries many of the elements of agonistic participation, deliberation and learning that are necessary in unstructured problem situations where there is little agreement on both the norms and values at stake and the type of knowledge required to solve the problem (Hoppe et al. 2013).

#### 4 Discussion

The case of the shrinking lakes thus demonstrates how water issues are becoming a public matter of concern and how people are getting involved in discussions about water management and governance. In these approaches, people identified many

challenges that are linked to all aspects of their daily lives in the area. These lives are intimately connected to the lakes but also to water more generally: social relations are created and shaped through water, economic factors impact decision-making on water governance, and private and public interests have to be negotiated alongside it (Krause and Strang 2016). New pressing water issues create new water publics, which raises questions about sustainable water distribution and the governance of water in a private and common approach (McDonald 2018).

Skepticism toward visitors and climate change impacted both the perceived challenges and the resulting perceptions of actions to address these challenges. First, in line with the postulations of positional economics, residents were concerned that the increased number of visitors produced or exacerbated these challenges due to the overuse and degradation of limited environmental amenities. This aligns with the idea that competition over positional goods (i.e., access to aesthetically pleasing and recreationally important landscapes due to proximity) can generate social tension (Hirsch, 1976). Despite such social tension and the emergence of citizen initiatives to address these challenges, skepticism toward climate change may limit individual behavioral changes that could improve outcomes for the lake, such as reducing private water consumption or carbon emission intensive behaviors. A large majority of residents perceive climate change to be an issue for the lake, and most residents would reduce private water consumption. A minority of residents remains skeptical about climate change. Correspondingly, the results of the choice experiment indicated a reduced willingness to pay for various improvements at Groß Glienicker Lake with increasing skepticism. Although the translation of environmental concern into behavior may depend on low costs (Diekmann et al., 2003), differing perceptions of responsibility for the causes of lake level loss as well as for the improvements at the lake may skew the willingness to act. Diekmann and Faist (2025), drawing on the "Imperative of Responsibility" framework from Hans Jonas, also found that perceiving climate change as a threat and assuming responsibility for current impacts were significant predictors of environmental responsibility toward future generations.

Through our mixed-methods approach, it became apparent that residents perceive the distribution of responsibility for the socionatural challenges differently. With governance structures fragmented, personal actions emanate from assuming personal responsibility, such as reducing water consumption, and collective actions also emerge such as the citizen initiatives. This insight addresses at least one condition of the recently proposed motivation-capacity-ownership framework for explaining why citizens join environmental initiatives in the energy sector. Self-efficacy (i.e., whether one believes in being capable of providing a public service) and response efficacy (i.e., whether one thinks actions are effective in delivering a service) are necessary for deciding to join such initiatives (Mees, 2022). Subsequently, further extensions on environmental psychology from group theory suggest that collective efficacy (i.e., believing that the group is capable of effecting change) can further promote feelings of self-efficacy (Jugert et al., 2016). The activities of the citizen initiatives and the joint network show the potential to activate agency through formation of groups. The use of this potential in combination with a potential non-hierarchical dialog structure between residents and representatives of administration, politics and science could increase the experience of self-efficacy among residents and enable new forms of collaboration. As income and education levels of stakeholders and residents indicate, they have the means and privilege to make efforts to enter debates. Apart from considering them as necessary participants on site, the current discussions and practices at the two lakes show possibilities of new forms

of political and scientific communication about the future of and at the lakes. The majority of residents and stakeholders want to have an informed discourse about possible actions, including both technical solutions and new forms of management and communication along the shores of the lakes. They value discussions in forums and platforms where they do not just want their wishes to be implemented, but where the different parties involved think through solutions together. This includes making political decision-making processes transparent and accessible to those who would like to act but have personal or economic constraints, or do not yet know how to contribute.

In light of this reasoning, discussions about the option spaces under consideration should not be limited to technical options for restoring lake levels. Rather, it is important to understand the social and emotional entanglements of the hazard as well as the socio-political structures that manifest in the hazard and to take them into account when discussing options for future action. Starting points for such options for climate adaptation "beyond technical fixes" (Nightingale et al., 2020) can be broadly conceptualized as instances of adaptive governance, a concept that emphasizes the need for flexible governance mechanisms capable of dealing with uncertainty and complexity in social-ecological systems that traditional top-down governance approaches often fail to address (Chaffin et al., 2014). Central aspects of adaptive governance involve utilizing feedback loops, interconnecting policy actors across multiple levels, and involving a broad diversity of perspectives and stakeholders (Visseren-Hamakers et al., 2021; Spencer and Alexandra, 2024).

Reflecting on our case of the decline in water levels in Groß Glienicker Lake and Sacrower Lake as a socionatural hazard, our analysis offers a few starting points for a more adaptive governance at the lakes. First, the governance challenges are in part caused by the fragmented administrative responsibilities, which, in the opinion of many stakeholders, contribute to the perceived inactivity or inability to decide and act on the part of the responsible administrative bodies. Second, the perceived lack of implementation of existing regulations and significant uncertainties regarding the planning of growing neighborhoods are reflected in the residents' desire for better enforcement of rules, for which the local authorities are held particularly responsible. Third, against this background, the willingness to act on the part of residents and the already existing local practices of dealing with the perceived socionatural challenges are particularly relevant with regard to the implications of our research and possible starting points for future action.

From an adaptive governance perspective, these local practices of citizens around the lakes and their relations to local administration and politics are valuable resources because they resemble the two classic roles of NGOs in environmental governance, which Jasanoff captioned as "criticism/reframing" and "epistemic networks". The first role, in Jasanoff's words, describes the "criticism of dominant scientific and policy frameworks (...) founded on (...) local environmental knowledge" (Jasanoff, 1997, p. 581). The citizen initiative campaigns are good examples of this, as they produce and present "localized knowledge of nature gained through non-scientific activities" (ibid., p. 583) to challenge the authority of both procedures and knowledge on which administrative (non-)decisions around the lakes are usually based. The second role, which is characterized by "creating more inclusive 'epistemic networks' around (...) defined environmental objectives" and facilitating "consensual action because of their experience in integrating environmental concerns with other aspects of community life" (ibid., p. 581), is represented both by the internal collaboration between the citizens' initiatives involved and by their cooperation with the

as their outreach to other community members represent "bonding and bridging processes" within the affected civil society, while cooperation with the authorities in the context of the dialogue represents "linking processes" between civil society and administration (van Dam et al., 2014, p. 336).8 Even though things are far from perfect in terms of the cooperation between citizens and local authorities, these processes of bonding, bridging and linking spurred by the local citizens' initiatives are indispensable for building good relations, developing trustful relationships and collaborating successfully on the local level (see also Hassink et al., 2016). To improve this, the existing governance challenges and the current lack of adaptive governance referred to above should be recognized and be addressed, both through transparent communication and by creating options for controversial discussions and participation. Such a holistic approach would ensure that local problems such as the lake level decline as well as underlying challenges such as climate change, demographic change and lacking government capacities can be dealt with in the best possible way. Our mixed-methods approach provided a nuanced understanding through different perspectives, but a few limitations do need to be mentioned. Despite the three different research approaches, it is possible that we were not able to garner all perceptions. The survey administration did not follow a systematic and random sampling approach, making the extrapolation of sample results to the population level difficult. In addition, online surveys inherently embed a potential sampling bias, although the results of the survey were evaluated in collaboration with insights from the ethnographic work with interviews, observant participation in events at site, and analysis of current political actions from local initiatives. Statements of perception can never stand alone but were discussed in relation to practices of people, the results of the survey and insights from workshops. Furthermore, although the stakeholder workshops included many representatives, it is possible that only interested parties participated in the workshop. Regarding both the ethnographic research and the stakeholder workshops, the interviewer bias could have led to potential issues; however, we managed this through joint interviews in initial stages of the research. Also, the potential influence of subjectivity on the results was further managed through transparency, mutual participation to different degrees by the researchers in the different approaches and active and iterative communication and collaboration in

authorities in the context of the citizen dialogue. The internal collaboration between the citizens' initiatives involved as well

the elicitation of findings.

<sup>0</sup> 

<sup>&</sup>lt;sup>8</sup> Van Dam et al. describe bonding as building "trusting co-operative relations between members of a network who are similar in terms of social identity", while bridging refers to building "connections between those who are unlike each other yet are 'more or less equal in terms of their status and power", and linking refers to connecting "individuals and groups in different social strata in a hierarchy where power, social status and wealth are accessed by different groups. (...) Applied to the practice of citizens' initiatives, the process of bonding refers to the interaction between the initiators and other local groups with different interests or orientation such as farmers, entrepreneurs, local residents who go back generations and more recent arrivals; and the process of linking refers to the interaction between initiators and institutional actors" (van Dam et al., 2014, p. 326).

## 5 Conclusion

perceived as a serious challenge by local residents and other stakeholders. The shrinking lakes have become a growing concern, shaping perceptions and initiating further local action while also raising uncertainties. Our mixed method approach garnered insights across overarching themes including (1) perceptions of socionatural change, (2) the social structures that interact with these perceptions, (3) the willingness to act and perceptions of responsibility, and (4) local practices for dealing with the challenges. Based on this new attention to the lakes, topics such as environmental protection and water management, but also issues of social and economic infrastructure and community care, have gained a new presence for the people and are perceived as challenges. These challenges are related to and influenced by social structures and a political history of division that have had an impact on the different private and public approaches to the lake level loss. Attention to this issue has led to motivation to act, as well as uncertainty and a desire for greater public control. This, in turn, is connected to the perception of a fragmented governance structure perceived to cause delays in local efforts to deal with the problem. This has brought with it doubts about public and private responsibilities and the local people's perception of self-efficacy. Nonetheless, we have shown that these perceptions and doubts have not dampened the dedication and commitment of local citizen initiatives to take matters into their own hands and influence others. On the contrary, we found a diversity of local practices of concerned citizens already addressing the perceived challenges as well as instances of collaboration between them and local authorities, which indicate that concerted efforts on the part of the former are in fact appreciated by the latter, but also that responsibilities should be more clearly distributed between citizens, administrative bodies, and political decisionmakers. The overall objective of this study was to understand lake level loss of Groß Glienicker Lake and Sacrower Lake as a hazard through its socionatural relations and how these relations in turn frame discussions about the hazard and how to deal with it. We therefore conceptualize and understand the decline in lake water levels as a socionatural hazard, where change in the biophysical environment causes material or physical harm as well as social, political and emotional consequences, Based on

Over the course of this paper, we have shown how the declining water levels in Groß Glienicker Lake and Sacrower Lake are

through its socionatural relations and how these relations in turn frame discussions about the hazard and how to deal with it. We therefore conceptualize and understand the decline in lake water levels as a *socionatural* hazard, where change in the biophysical environment causes material or physical harm as well as social, political and emotional consequences. Based on an interdisciplinary social science mixed-methods approach, this expanded understanding of lake level loss as a *socionatural* hazard offered a prism through which a broader perspective allowed us to reveal how social dynamics and water hazards mutually constitute each other. We captured a full range of local and collective perceptions of the current hazard in a challenging context of diverse administrative jurisdictions and complex scientific investigations and to understand how and why local actors (still or precisely because of the complexities) feel responsible and inclined to act in such circumstances, thereby also informing local adaptive risk governance. Thus, this paper contributes to a broadened conceptualization of natural hazards as *socionatural* hazards and, in addition, shows that this endeavor is not purely academic, but also has concrete practical implications.

The manner in which society discusses and governs water resources is undergoing substantial transformations, driven not only by climate change but also by a multitude of challenges, especially at the local level. The case of Groß Glienicker Lake and

- Sacrower Lake in Berlin-Brandenburg demonstrates how the impacts of these changes vary depending on the social and
- hydrological conditions present, which can be highly heterogeneous even within a relatively small geographical area. These
- contrasts in local contexts, from differing risk perceptions of water shortage to distinct governance needs, not only underscore
- the necessity of adaptive governance approaches that are fine-tuned to the specific hydrological systems in question. They also
- reinforce calls to rethink current governance strategies and thus also how we frame our conversations about water. It is
- imperative that governance strategies are adaptive not only to nature-related risks but also to the diverse social functions, needs
- and challenges these systems represent, i.e., to their *socionatural* character and dynamics.
- Our study has provided a novel view of hazards beyond the physical aspects of water level decline by adopting an
- interdisciplinary approach to the study of hazards as socionatural hazards. Furthermore, it harnessed the potential of
- transdisciplinary research by discussing the complexities of governance efforts with the public, thereby also highlighting the
- constraints of such approaches, particularly in the context of addressing the more profound and transformative changes
- required for long-term adaptation. Nevertheless, recognizing and accounting for the heterogeneity of local risks and
- perceptions will be pivotal in developing more effective and responsive risk management models and overall successful
- adaptive governance approaches
- **Author contribution:** DH performed ethnographic research and analysis. DJ, JH, LM and TF conceptualized the household
- survey. DJ and JH administered the survey, and DJ performed the analyses. TV and PF undertook the stakeholder workshops
- and analysis. DH, TV and DJ equally contributed to the overall analysis and writing of the manuscript. JN, JH, and LM
- contributed with conceptualization, review and editing.
- **Competing interests:** The authors declare that they have no conflict of interest.
- **Special issue statement:** This article is part of the special issue "Current and future water-related risks in the Berlin–
- Brandenburg region". It is not associated with a conference.
- Acknowledgements: This research was funded through the Einstein Research Unit 'Climate and Water under Change' from
- the Einstein Foundation Berlin and Berlin University Alliance (ERU-2020-609).

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
