# Peer review of "Shrinking Lakes, Growing Concerns: Exploring Perceptions of Lake"

_EGUsphere, 2025_

## Author Comment (AC1)

**Responses to Reviewer 1 (EGUSPHERE-2025-475):**

| | |
|---|---|
| **RC1** | |

| | | |
|---|---|---|
| 1.01 | **Reviewer comment** | |
| | The paper is well written and focused. The results should be further developed and discussed on the basis of other socio-hydrological studies (on collective memory, decline of civilizations, governance and conflict over water use). Studies on water-scarce regions can help in the perspective of changing human desires. In particular, I would welcome more descriptive papers, but I must say that the results of this study are of local importance (the editor should take this into account). | |
| | **Author response** | |
| | Thank you for this valuable comment. We want to highlight here that all line numbers mentioned in our comments refer to the preprint version of our paper, and proposed changes to the manuscript are written in red font. | |
| | In this paper, we have focused on a local example that highlights the combined challenge of political complexity and specific natural conditions. For these reasons, we have refrained from comparing other regions, also to allow more space for bringing together our studies. We also welcome further papers on this topic, but this paper is in dialogue with other contributions to the special issue and contributes to a more nuanced understanding of the term "hazard" for the region Berlin-Brandenburg. To make this more explicit we will add this to our introduction. | |

| | | |
|---|---|---|
| 1.02 | **Reviewer comment** | |
| | How was the subjectivity of the researchers managed during the qualitative analysis? (add in the text). | |
| | **Author response** | |
| | Thank you for this question. Although not explicitly stated in the submitted manuscript, we did in fact address subjectivity through the use of different research methods. As highlighted below in our other responses, triangulation was not the primary goal of the study, but our approach does resemble triangulation in some steps of the research and also allows us to limit the degree of influence of subjectivity on the results. Therefore, we will incorporate an updated description in the 2.2 Mixed-methods approach section under line 180 as follows: "To fully exploit the potential of a mixed-methods approach, we considered the empirical data from all three studies and combined their findings to gain a deeper understanding of the mechanisms behind different local perceptions. Through this combination, we managed the potential influence of subjectivity on the results as the researchers maintained transparency, participated to different degrees in each of the different methodological approaches and actively and iteratively communicated and collaborated on the elicitation of findings across the research methods. Based on the empirical findings, we generated overarching themes, which we formulated as questions in order to relate results from individual empirical approaches to each other and to the overarching discussions in the joint research project." | |

| 1.03 | **Reviewer comment** |
|---|---|
| | Line 29 - check the concepts of hard-path to soft-path solutions (I think it would be interesting to add it to the work or quote it) |
| | **Author response** |
| | Thank you for this suggestion. We agree that the hard-path vs. soft-path distinction in water management is a valuable conceptual lens that aligns well with our findings. We will add this and make a brief reference to this framework towards the end of the relevant section of the manuscript (the paragraph starting in line 76) to better situate our analysis within existing debates on technical versus integrated approaches to water-related challenges. |
| | Reference to be added: |
| | Gleick, P. H. (2003). Global freshwater resources: soft-path solutions for the 21st century. Science, 302(5650), 1524–1528. https://doi.org/10.1126/science.1089967 |

| 1.04 | **Reviewer comment** |
|---|---|
| | Add legend to Figure 1. |
| | **Author response** |
| | Thanks for this suggestion. We will add a legend to figure 1 during the revision. |

| 1.05 | **Reviewer comment** |
|---|---|
| | Line 107 - I missed historical information and quotes about the lakes (add); |
| | **Author response** |
| | Although we appreciate this comment, we fail to see how further historical information and quotes about the lake could provide pivotal beneficial information beyond the current descriptions in the manuscript. We have already devoted a significant amount of space in lines 108-129 to describe the study area, including a description of the historical division of the study area between East and West Germany. Moreover, subsections 3.2.2, 3.2.3 and 3.2.4 (starting at line 309) describes the political history of the area. Lastly, the whole manuscript is devoted to better understanding the connections between the people and the lakes. We would propose to move some parts from sections 3.2.2, 3.2.3 and 3.2.4 to the 2.1 case study description and additionally to indicate that further information on the social structure, political history and administrative fragmentation of the lakes/of the area can be found in the corresponding sections (also see response to comment 2.07). |

| 1.06 | **Reviewer comment** |
|---|---|
| | Line 170 - is it possible to show this socio-hydrological system in a figure/loop/map? |
| | **Author response** |
| | We certainly agree that visualizations can be beneficial for understanding the system as a whole, but we do not see a possibility to represent the entire socio-hydrological system in |

one figure. Some visualizations were used during the stakeholder workshops, but these did not represent the system as a whole and would then make the understanding for the manuscript difficult. Therefore, we have to refrain from visualizing the socio-hydrological system, unfortunately.

| | |
|---|---|
| 1.07 | **Reviewer comment** |
| | Line 171 - what are the eight functions? |
| | **Author response** |
| | The eight functions are based on the common ecosystem services that have been customized for the case study together with workshop participants. This has been presented in Figure 6.: create an attractive and healthy living space and enable social participation, protect biodiversity and climate, conserve natural resources, regulate the water balance, preserve the historical-cultural identity of the region, ensure water-sensitive urban design, secure economic livelihoods, provide provisioning services. We will clarify this at the end of the sentence in line 172 with a reference to subsection 3.2.1 as follows: |
| | "The first step in this process was to define the study area as such a social-hydrological system based on its main physical and social characteristics, the core and contextual actors that shape it, and the essential functions it is supposed to fulfill (see figure 6 in section 3.2.1)." |

| | |
|---|---|
| 1.08 | **Reviewer comment** |
| | Methodology - perhaps it would be interesting to show the paths through a flowchart; |
| | **Author response** |
| | Thank you for this suggestion, and we agree with the idea to include a visual depiction of the work in the manuscript and how we arrived at the overarching themes and questions. We would propose the following visualization to be included at the end of section 2.2 Mixed-methods approach: |
| |
[Figure]
 |

| 1.09 | **Reviewer comment** |
|---|---|
| | Line 199 - it's worth noting that more than 40% answered that the quality of the water remained the same or they didn't identify anything, a percentage greater than the sum of a little and a lot; |
| | **Author response** |
| | Thank you for noticing this. We will adjust the description to better reflect the higher percentage of respondents noticing no change in the quality of the water. At line 199, the sentence will read: "Although 35 % of respondents felt that water quality 200 had declined slightly or greatly, and similar responses were received for bird, insect, and plant diversity, approximately half of the respondents either did not notice a change, or indicated that it had stayed the same." |

| 1.10 | **Reviewer comment** |
|---|---|
| | Line 203 - why have visitors increased? |
| | **Author response** |
| | The causes of the increase in visitors was not investigated in the course of the study, but perceptions in the interviews and the survey indicated that residents believe that the lockdowns during the coronavirus pandemic were associated with the increase. We highlighted this in line 335. Nonetheless, we will mention this already in line 203 in the revision as follows: |
| | "Strong majorities perceived that the number of visitors (> 80 %), vehicles (> 80 %), and litter (65 %) had increased slightly or greatly. Indeed, many open-ended responses highlighted the issues of littering, visitor pressure resulting from lockdown measures during the corona pandemic and also leading to trampling of shoreline vegetation, unsanitary infrastructure and wild bathing." |

| 1.11 | **Reviewer comment** |
|---|---|
| | Were there differences in perception between different groups of interviewees, such as old residents versus new residents or people with different levels of education? |
| | **Author response** |
| | This region is very dynamic, mainly due to demographic change since 1990 and further densification due to increasing pressure in the city of Berlin. Statements concerning the lakes cannot be separated according to age or educational aspects. However, there are varying degrees of concern depending on how close the actual place of residence is to the lakes. We have illustrated this using the results of the survey throughout section 3. |

| 1.12 | **Reviewer comment** |
|---|---|
| | Line 210 - characterize hydrology with historical data for the region (precipitation, evaporation, etc.) This could be added to the "study area" item; |
| | **Author response** |
| | We cannot to delve deeply into this, because we do not want to juxtapose the citizens' perceptions with "the truth", but we will add the following sentence on this in the case study area section (line 109): "Both lakes are exclusively fed by groundwater, i.e. they have no surface inflow and thus ultimately depend on the rainfall and the evapotranspiration in the case study area. As in the entire Berlin-Brandenburg region, the hydrology of the case study area is fundamentally characterised by dryness. While the annual precipitation remained more or less constant in recent decades, however, the spatio-temporal variability of precipitation patterns and in particular the changes in the frequency and intensity of precipitation extremes have increased significantly in the context of climate change, as historical data show (Bart et al., 2025)."

Reference to be added:

Bart, F., Schmidt, B., Wang, X., Holtmann, A., Meier, F., Otto, M., and Scherer, D.: The Central Europe Refined analysis version 2 (CER v2): evaluating three decades of high-resolution precipitation data for the Berlin-Brandenburg metropolitan region, metz, 33, 339–363, https://doi.org/10.1127/metz/2024/1233, 2025. |

| 1.13 | **Reviewer comment** |
|---|---|
| | Line 256/257 - are there any federal/state/municipal public policies on land use and occupation or master plans? |
| | **Author response** |
| | There are a plethora of federal/state/municipal public policies and strategies/master plans that affect land use and occupation in the region. On the one hand, though, it is impossible to represent them in their entirety. On the other hand, it is not possible to pick one or a few representative ones. On a more abstract level, Dombrowsky et al. (2022) address the aforementioned governance-related issues, albeit not entirely, and the solution remains somewhat beside the point. As a result, we cannot satisfy this request, because doing so would be disproportionate in the context of this paper, both in terms of the effort needed and the space it would take up in the paper.

Dombrowsky, I., Lenschow, A., Meergans, F., Schütze, N., Lukat, E., Stein, U., & Yousefi, A. (2022). Effects of policy and functional (in) coherence on coordination–A comparative analysis of cross-sectoral water management problems. Environmental Science & Policy, 131, 118-127. |

| 1.14 | **Reviewer comment** |
|---|---|
| | Line 297 - it would be interesting to present the number of cafés, restaurants and parking spaces in the area. The density of these services can influence the social perspective; |

| | **Author response** |
|---|---|
| | Thank you, this is a valid point. We will add a short description following line 280 as follows:

"When stakeholders were asked about the essential functions of the lake as a social-hydrological system during the workshops 280 (see section 2.2), an attractive and healthy living space offered by the lakes and the social participation they enable were rated

highest (see Fig. 6).2 The residential area of Sacrow, Groß Glienicke and Kladow is currently characterised by its proximity to the city, but its distance from the urban centre gives it a village feel. The necessary shopping facilities are complemented by a few cafés and a handful of restaurants. A restaurant on the northern shore of Lake Sacrow has been converted into a temporary event venue, and the café on Lake Groß Glienicke is usually open on weekends during the summer months. Both residents and visitors emphasise the tranquillity of the area and the natural surroundings by the water. The preference to keep this healthy environment intact is thus related to the rising number of people and visitors and their (allegedly) harmful behavior, which is perceived as a challenge or threat (see previous section). Closely related to the attractiveness of living space and social participation is the environmental health of the system, represented by the functions of biodiversity and climate protection as well as the conservation of natural resources, which are perceived as being challenged indirectly by the increasing 285 number of users and their harmful behavior, but also - more importantly - directly by the decline in lake water levels and quality. Lastly, the meaning of the lakes with regard to the preservation of the historical and cultural identity of the region plays a meaningful role in the participants' evaluation of the essential functions of the social-hydrological system." |

| 1.15 | **Reviewer comment** |
|---|---|
| | Line 409 - I believe there may be a relationship with collective memory here. Please, if possible, link to this theme; |
| | **Author response** |
| | This is a valid point, and thank you for pointing this out. This is certainly very important when it comes to seeing the lake as a memorial of German history. However, since this area is highly dynamic and changed considerably since 1990 to apply collective memory here would need further work with the material. We will pick this up in future contributions. At this point it would take the paper into a different direction. |

| 1.16 | **Reviewer comment** |
|---|---|
| | Line 427 - Is it possible to make a loop figure with these identified interactions? |
| | **Author response** |
| | The results discussed in line 427 refer to the analysis of the choice modeling. This modeling associated choices made in the discrete choice experiment with the respondents' answers to the climate change skepsis questions. Therefore, eliciting these one-way interactions in a loop figure would not be appropriate or elicit any further understanding of the interactions. |

| 1.17 | **Reviewer comment** |
|---|---|
| | Line 501 - I think the political bias could be explored further. How can the political spectrum influence the conservation policies of these lakes and the environment (for example, the city being run by a left-wing or right-wing politician)? |
| | **Author response** |
| | It is a little unclear to us what is meant by "political bias" here. The political nature of the actions of citizens with regard to the lakes that is mentioned here is to be understood as political in a very basic sense, i.e. relating to the public affairs of a community and challenging the respective (non-)decisions and (non-)actions of the involved governments/administrations. Party politics, in fact, play a very subordinate role in this respect, i.e. with regard to such a very specific local issue like the one at hand here, as one interview specifically said. |

| 1.18 | **Reviewer comment** |
|---|---|
| | Line 625 - Is it possible to add a paragraph of recommendations for the population, public administration or other researchers? |
| | **Author response** |
| | Thank you for the suggestion to include recommendations. Our study was focused on understanding the hazards and perceptions around the two lakes rather than directly looking for solutions to address the hazards. Therefore, we cannot jump to recommendations for the population, public administration or other researchers in taking care of the hazard. Nonetheless the conclusion starting at line 625 does shed light on how understanding the diverse social challenges and functions are important and pursuing approaches to elicit these allows for a broader understanding of the socionatural nature of hazard beyond the declining lake levels as a natural hazard. Given these conclusions in light of our research questions, we refrain from providing recommendations beyond methodological approaches for understanding hazards. |

---

## Author Comment (AC2)

**Responses to Reviewer 2 (EGUSPHERE-2025-475):**

| RC2 | |
|---|---|
| 2.01 | **Reviewer comment** |
| | The paper addresses an important issue of how local residents perceive the impact of climate change and what kind of actions they would prefer to cope with the present challenges. |
| | The paper uses a triangulation of methods (without using the word triangulation which was surprising to me) combining quantitative surveys, qualitative interviews, and stakeholder workshop results. In addition, they processed official documents of several water authorities. |
| | **Author response** |
| | Thank you for this valuable comment. We want to highlight here that all line numbers mentioned in our comments refer to the preprint version of our paper, and proposed changes to the manuscript are written in red font. |
| | In line with our approach, which is not to validate the results of our studies but rather to "deepening and widening" our understanding, we added the following sentence in line 181: "We thus used the method of triangulation, although not primarily to cross-validate our empirical findings, but rather to deepen and widen our understanding of the socionatural phenomena at and around the lakes in the sense of Yeasmin and Rahman (2012)." |
| | Reference to be added: |
| | Yeasmin, S. and Rahman, K. F.: 'Triangulation' Research Method as the Tool of Social Science Research, BUP JOURNAL, 1, 154–163, 2012. |

| 2.02 | **Reviewer comment** |
|---|---|
| | The introduction provides a clear and convincing argument for using quantitative and qualitative methods for assessing the public responses to a dramatic decline in water level associated with climate change. However, the introduction is quite long and often redundant stating the main messages several times. |
| | **Author response** |
| | We agree that the introduction is rather long, but we believe this is justified given the dense and complex subject matter considered in the paper as well as the diversity of research methods used. However, we would agree to make minor adjustments to the introduction to improve the readability and prevent repetition of the main messages. |

| 2.03 | **Reviewer comment** |
|---|---|
| | The theoretical concept of the study remains fuzzy. The authors seem to base their arguments on the social constructivism school of thought but oscillate between framing the issue in terms of "real" problems and "perceived" problems (Thomas Theorem). Some |

expressions in the text are complete nonsense: "In this context, the declared hazard of lake level loss shapes and is shaped by perceptions". This sounds like magic: the water level can certainly not be changed by perceptions. What the authors probably try to say is that the assessment of water decline depends on how observers perceive and evaluate the physical changes that they observe.

**Author response**

Thank you for raising the concern about the fuzziness of the theoretical concept. You are exactly right in your interpretation and we will make changes in the introduction to improve the clarity of the frame of analysis.

What we are investigating here are the perceptions of the environment that generate a discourse of ''hazard.'' It is only through politicized human-environment interaction that the sinking water level is perceived as a hazard. This discourse entails certain practices, which in turn influence the perception of the lake and its changes. As the reviewer points out, we want to challenge an objective perception and at the same time show that the hazard is much more than an environmental change that then affects social processes, but must be understood as a common, socionatural process. That does, of course, not mean that the physical water level is changed by the human perceptions of said water level. We will further clarify our approach in the introduction and adjust the wording and explanations in the sections mentioned and throughout the manuscript.

| 2.04 | **Reviewer comment** |
|------|----------------------|

Later on, they introduce the concept of socio-natural phenomena which emphasize the coupling between natural dynamics and human perception. There is a vast tradition in the social sciences using the concept of socio-ecological systems which is based on the same line of thinking, but this popular concept is not mentioned in the paper.

**Author response**

We appreciate the comment about the lack of clarity about the theoretical underpinnings. We agree that there is a rigorous scientific basis for social-ecological hazards and systems. This line of understanding focuses on how social and environmental factors interact and may be at risk in the context of the vulnerability to natural hazards. We have chosen to consider socionatural hazards because of a slightly different focus and scope, given that speaking of socionatural hazards emphasizes the inseparability of nature and society when it comes to "natural" hazards. In doing so, we examine not only the interactions between human society and the natural environment in relation to the hazard of lake level loss, but also how the hazard itself is defined through the perceptions of people, including those living in and around the lakes, as well as other stakeholders.

We will conduct a thorough review of the paper as a whole with a view to conceptual coherence and add a clarifying sentence in the introduction (also related to the previous comment): ""Socio-ecological" approaches treat society and nature as interacting but separate systems. In contrast, "socionatural" perspectives emphasize their inseparability—seeing nature as socially produced and society as materially embedded. In this paper, we therefore adopt a socionatural perspective in order to avoid reifying the nature/society

divide and to better grasp and understand the relationality and dynamics of human–nature entanglements (West et al., 2020)."

Reference to be added:

West, S., Haider, L.J., Stålhammar, S., Woroniecki, S., 2020. A relational turn for sustainability science? Relational thinking, leverage points and transformations. Ecosystems and People 16 (1), 304–325.

| 2.05 | **Reviewer comment** |
| --- | --- |
| | The methodology section includes an impressive list of research methods that have been used to assess attitudes, values, traditions and policy preferences. An interdisciplinary array of research traditions has been included in the study. (ethnographic, survey, stakeholder workshops) However, it was not quite clear to me which of the three research questions was addressed by what methods and how the results were integrated. |
| | **Author response** |
| | Thank you for the assessment of our research methods. In many cases, the boundaries between the research methods were fluid in that findings from one method helped to directly interpret results from another method. Therefore, we did not structure the results section to explicitly separate the results among the three methods. This may make it difficult for readers to understand which questions were addressed by which methods. However, we have described our integration process in the methods section (i.e., starting on line 180). Not all research methods were used to address all thematic blocks, and more than one research method was always used in a thematic block. In addition, we have described the connections between the research methods through the description of the results, such as in lines 240-245 where all three research methods are cited, or at the beginning of many paragraphs (e.g. in line 264 describing both the survey and stakeholder workshops). |
| | In order to make it easier to understand which main results are derived from which research method, we will make improvements in several areas of the results section, as described below: |
| | Line 193: People living around the lakes are very aware of the obvious decrease in water. In initial conversations during ethnographic research on the street, in interviews around kitchen tables and at events, the lack of water was the first thing to be emphasised. This was not only the starting point for conversations initiated by the researchers, but also the linchpin of the lake residents' own problem analysis. This local problem analysis has created a public that sees itself as divided between people who say that they are very concerned about what is happening in their environment and about the causes and possible solutions that are being developed in political arenas, who ask questions and build up political pressure, and residents who see the problem but feel unable to do anything about it. In general, discussions and practices start with this narrative of the danger of the sinking lake level. However, the challenges associated with this first narrative point to a more complex picture of this hazard. |
| | Line 200: "During the joint exploratory phase with joint interviews to prepare the survey questions, the topic of lake level loss very quickly opened up discussions about further |

problems of everyday life in this area, putting into context the other challenges mentioned in close relation to the falling lake level (also in the initial interviews with residents)."

Line 203: "In the survey, strong majorities perceived that the number of visitors (> 80 %), vehicles (> 80 %), and litter (65 %) had increased slightly or greatly."

Line 207: "Figure 2. Perceived awareness of changes at Groß Glienicker Lake from the survey (N = 644)."

Line 213: "Further responses in the interviews indicated a certain ecological understanding and means of anthropogenic impact among the residents, how important the water level as well as the natural vegetation along the shore are for maintaining good water quality. Here, residents see pressures from the population and rising number of visitors and highlighted the ecological challenges with people not taking care of their surroundings (i.e., trampling of reed vegetation along shore by the increasing number of visitors). The ecosystem was one of the main concerns expressed in conversations. However, these are weighed against the utility value that residents in this region derive from them. This leads them to develop an appreciation for their surroundings, which they then prioritise preserving. We will go into further detail about this in section 3.4."

Line 221: "Interestingly, the risks perceived expressed in the survey that are associated with climate-related changes may not be consistent with the overall perception of climate change risks in the region."

Line 220: "Figure 3. Perceived challenges or dangers for Groß Glienicker Lake from the survey (N = 644)."

Line 232: "Figure 4. Climate change skepticism among residents from the survey (N = 644)."

Line 266: Participant observation revealed that both lakes are open to the public in places, and restricted in others, either by private ownership or environmental protection zones …

Line 290: "Figure 6: Ranking of essential functions of the social-hydrological system of the lakes from the stakeholder workshops (N = 8)."

Line 300: "Figure 7. Preferences for future changes at Groß Glienicker Lake from the survey (N = 644)."

Line 341: The demographic development and social structure of the region has developed from a rural area with a settlement history that dates back to the 13th century, to a village structure mixed with middle-class city dwellers at the beginning of the 1920s, and then has been shaped by separation and finally German reunification. This history has been raised both in interviews as well as in workshop group discussions.

Line 347: "As made apparent in the ethnographic research, this had a significant impact on everyday life surrounding the lake, especially on the geographically western (politically eastern) side of the lake as the wall cut off Groß Glienicke residents from access to the lake who had to go to Sacrower Lake instead."

Line 417: "Figure 8. Indicated willingness to contribute among the respondents to the improvement of the situation at Groß Glienicker Lake from the survey."

Line 431: "Table 1. Willingness to pay (WTP) estimated through the conditional logit model including interaction terms from the survey."

Line 450: "Figure 9. Perception of responsibility by the residents concerning who should implement improvements at the Groß Glienicke Lake from the survey."

Line 461: "Despite its relatively small size and population, the ethnographic research revealed a flurry of citizens' initiatives in the three residential areas around the lakes concerned with the lakes and their ecosystems."

Line 457: In this context, while some ask themselves what their contribution could be, others got active and organised themselves in several initiatives. How do the different actor groups, especially the latter, practically deal with the perceived challenges and responsibilities? In the following we refer to citizen initiatives we worked with in workshops and met during site visits and their practices in dealing with perceived challenges connected to the declining water levels.

Line 459: In the following we refer to citizen initiatives we worked with in workshops and met during site visits and their practices in dealing with perceived challenges connected to the declining water levels.

| 2.06 | **Reviewer comment** |
| --- | --- |
| | The results section starts with an overview of what the authors learned about perception. This section is highly dominated by the survey results while the results of the qualitative interviews seems to be almost lost; at least the striking advantage of triangulation combining proportional results from quantitative data with more in-depth-results from the interviews was not adequately delivered. |
| | **Author response** |
| | Thank you for this comment. We will provide a more detailed presentation of the contributions from the studies in the relevant sections (see also the previous comment), so that the contribution of the different empirical approaches, especially of the ethnographic fieldwork, as well as their combination will become clearer. |

| 2.07 | **Reviewer comment** |
| --- | --- |
| | The second section on social structures was informative but confusing in terms of organization. Results from the surveys were mixed with background information on the politics and history of the area (which definitely belongs into section 2.1: Case study description), It also seems quite awkward to switch from perceptions to social history, to administrative governance and back to preferences. All information that is being presented is valuable, but it would need re-organization. I would place the history, the administrative arrangements and the social structure of the residents in the Case Study description and focuses on the empirical results in section 3.2 and 3.3. The description of the discrete choice experiment seemed like an add-on to the study and the relationship to the other results were only briefly mentioned on a rather superficial level. |
| | **Author response** |
| | We agree with the reviewer that some of the information in section 3.2 describes the case study situation. However, several parts of Section 3.2 also describe empirical results |

obtained through the three different research methods. To provide a coherent and fluid depiction, we opted to keep the topics in subsections 3.2.2, 3.2.3 and 3.2.4 together in the results section, thus avoiding the need for continuous back-references to the case study description. Therefore, we propose maintaining the original structure to provide readers with a concise and coherent description. However, we would propose to more clearly differentiate between purely descriptive and empirical/analytical parts and therefore move some parts from sections 3.2.2, 3.2.3 and 3.2.4 – especially the ones that are clearly descriptive and of no or little analytical value in relation to our empirical work – to the 2.1 case study description. Additionally, we will indicate that further information on the social structure, political history and administrative fragmentation of the lakes/of the area can be found in the corresponding sections (also see response to comment 1.05) and make minor adjustments throughout the text to make it clear where the information stems from.

Regarding section 3.3, the discrete choice experiment was an equally important method in the research to understand the willingness to act in the form of a financial contribution and how such preferences might change in light of differences in climate change skepticism. To provide a better indication of how the choice experiment results tie into the wider insights of the paper, the discussion will be updated as described in comment 2.08.

| 2.08 | **Reviewer comment** |
|---|---|
| | The discussion highlights the main results and tries to infer some major messages from the mixed method approach. This is well done in my view although, similar to the introduction, quite redundant and wordy. What I miss was a comparison of the results with similar studies or interpretative frameworks. The insights that residents are skeptical about newcomers which may challenge their privileges by overusing the finite pool of resources is a common theme in positional economics (F. Hirsh) or group theory in social psychology. The preference for low-cost policies is well documented in the literature on climate change adaptation (for example A. Diekmann). I would advise the authors to add a section on how their results fit into the present status of knowledge about the phenomena that they describe. Where do they confirm what is already known and where do they add something new? |
| | **Author response** |
| | We appreciate the suggestion to better compare the results with similar studies. We have supported our findings in the discussion section with several references to the literature, but we will provide a further basis in a paragraph of the discussion to reflect the skepticism of tourism and climate change starting at line 569: |
| | "Several of the main findings of our approach also reflect insights gained from other areas of literature in different contexts. For instance, skepticism toward tourists and towards climate change impacted both the perceived challenges and the resulting perceptions of actions to address these challenges. First, in line with the postulations of positional economics, residents were concerned that increased tourism produced or exacerbated these challenges due to the overuse and degradation of limited environmental amenities. This aligns with the idea that competition over positional goods (i.e., access to aesthetically pleasing and recreationally important landscapes due to proximity) can generate social |

tension (Hirsch, 1976). Despite such social tension and the emergence of citizen initiatives to address these challenges, skepticism toward climate change may limit individual behavioral changes that could improve outcomes. Although the translation of environmental concern into behavior may depend on low costs (Diekmann et al., 2003), the survey results revealed a low, but very apparent, level of climate change skepticism among the sample population, with increasing skepticism being associated with a reduced willingness to pay for various improvements at Groß Glienicker Lake. Diekmann and Faist (2025), drawing on the "Imperative of Responsibility" framework from Hans Jonas, also found that perceiving climate change as a threat and assuming responsibility for current impacts were significant predictors of environmental responsibility toward future generations. Through our mixed-methods approach, it became apparent that residents perceive responsibility for the socionatural challenges differently, and that while governance structures are fragmented, personal actions may not always emanate from personal responsibility, such as in reducing water consumption, but rather collective actions emerge such as the citizen initiatives. This insight addresses at least one condition of the recently proposed motivation-capacity-ownership framework for explaining why citizens join environmental initiatives in the energy sector in that not only is self-efficacy necessary (i.e. whether one believes in being capable of providing a public service) but also response efficacy (i.e. whether a one thinks actions are effective in delivering a service) for deciding to join such initiatives (Mees, 2022). Subsequently, further extensions on environmental psychology from group theory suggest that collective efficacy (i.e. believing that the group is capable of effecting change) can further promote feelings of self-efficacy (Jugert et al. 2016). In this light, it is possible that further success of citizen initiatives helps to promote the identification of self-efficacy and individual responsibility for actions to address the socionatural challenges at the lake."

Diekmann, I., & Faist, T. (2025). Does the future have a lobby? Environmental degradation and perceived environmental responsibility towards future generations. Environmental Sociology, 11(2), 259-272.

Hirsch, F. (1976). Social Limits to Growth. Cambridge, MA: Harvard University Press.

Jugert, P., Greenaway, K. H., Barth, M., Büchner, R., Eisentraut, S., & Fritsche, I. (2016). Collective efficacy increases pro-environmental intentions through increasing self-efficacy. Journal of Environmental Psychology, 48, 12-23.

Mees, H. L. (2022). Why do citizens engage in climate action? A comprehensive framework of individual conditions and a proposed research approach. Environmental Policy and Governance, 32(3), 167-178.

| 2.09 | **Reviewer comment** |
| --- | --- |
| | The conclusions are partially redundant with the discussion. However, if the discussion is more focused on comparing the results with the literature, the conclusions can be kept as they are. |
| | **Author response** |
| | Thank you for highlighting this. |

| | |
|---|---|
| | We will adjust the discussion as suggested in the previous comment, and we will leave the conclusion as it is (apart from the references to the four research questions mentioned in the introduction; see comment 3.25). |

| 2.10 | **Reviewer comment** |
|---|---|
| | The paper needs a thorough language editing (grammatical errors, misleading expressions, and awkward wording). |
| | **Author response** |
| | We will subject the paper to thorough language editing and, inter alia, make the following specific changes: |
| | Line 464: "For example, as one of the smallest parts of Potsdam, the remote Sacrow has a very active and influential citizenry for all matters relating to this neighborhood, and stakeholders underlined the importance of taking care of their lakes." |
| | Line 468: "Popular topics became, for example, the water transfer from rivers to lakes, using the lakes as water storage in order to provide freshwater for the metropolitan region." |
| | Line 470: "For immediate results, especially water use and water extraction by residents and the nearby waterworks were put in focus." |
| | Line 471: "Although private consumption is difficult to track, the stakeholders were surprised that figures from the waterworks were not available (although numbers show that these have a minor effect on water levels)." |
| | Line 571: "These lives are connected to the water: social relations are created through and over the water, economic factors influence decision-making, and private and common interests have to be negotiated alongside it (Krause and Strang 2016)." |
| | Line 621: "To improve this, the existing governance challenges and the current lack of adaptive governance referred to above should be recognized and be addressed, both through transparent communication and by creating options for controversial discussions and participation." |

| 2.11 | **Reviewer comment** |
|---|---|
| | In summary, the paper is a valuable contribution to the field of behavioral studies on climate change adaptation and includes an excellent combination of methods for reaching a convincing triangulation. However, it needs some revisions with respect to organization, precision and integration into the existing body of literature. |
| | **Author response** |
| | Thank you for the overall positive reception of our manuscript. Given the comments made also by the other reviewers concerning the organization, precision and integration, we hope you agree with the changes suggested in our other responses to improve the manuscript. |

---

## Author Comment (AC3)

**Responses to Reviewer 3 (EGUSPHERE-2025-475):**

| RC1 | |
|---|---|
| 3.01 | **Reviewer comment** |
| | The paper addresses relevant scientific questions within the scope of NHESS, contributing to the understanding of natural hazards and their societal impacts. It presents new data and novel insights, with methodologies that align with international standards. The methods and assumptions are valid, but the description should be improved to ensure greater transparency in the research process. The results are sufficient to support the research questions, and significant insights are drawn from the analysis. While the study provides an explanation of the data, methods, and results, the clarity could be improved to enhance the reproducibility of the research for fellow scientists. |
| | **Author response** |
| | Thank you for this valuable comment. We want to highlight here that all our comments refer to the preprint version of our paper. All changes are added to this document. We will address these points in the context of more specific comments regarding these issues (made above, by the other two reviewers, and below, by this reviewer). |

| 3.02 | **Reviewer comment** |
|---|---|
| | The title and abstract are well-aligned with the content of the paper, providing concise and accessible summaries of the research. The paper is well-structured, with quality figures. Overall, the presentation is clear, the technical language is precise, and the English is fluent and easy to understand for a diverse audience. |
| | **Author response** |
| | We appreciate the overall positive reception of our manuscript. |

| 3.03 | **Reviewer comment** |
|---|---|
| | The author's contributions are clearly outlined, however, the credit to prior work is insufficient. While the references are appropriate and accessible, greater acknowledgment of related research would strengthen the scientific merit of the paper. There are several claims regarding the past, current, and future status of the study area that are not supported by references. It is recommended to either provide appropriate sources for these claims (e.g., scientific literature, policy documents, local reports) or clearly describe in the Methods section how this information was obtained. |
| | **Author response** |
| | Thank you for bringing this to our attention. We will add some more literature on critical water management (Hüesker, Moss, Naumann 2012), multilevel water governance (Moss and Newig 2010), and contested areas of water use in the region (Thierauf et al 2024) in line 54 as part of the introduction. We will include a reference to the website of the citizen initiative for the paragraph starting line 43. |

| 3.04 | **Reviewer comment** |
|---|---|
| | The study results do not clearly distinguish between findings based on perceptions of current conditions and those concerning future expectations. It is recommended to clarify this distinction to improve the readability and interpretation of the results. |
| | **Author response** |
| | Thank you for this comment. We will address this issue in the further specific comments as in comments 3.14, 3.17, 3.18 and 3.19. |

| 3.05 | **Reviewer comment** |
|---|---|
| | The title refers to 'socionatural hazards,' whereas the term 'social-ecological hazards' is used in the Abstract section. To improve clarity and coherence, it would be helpful to clarify whether these terms are intended to be synonymous and ensure consistent use of terminology throughout the text. |
| | **Author response** |
| | Thank you for noticing the use of the two different terms. We will improve the clarity and coherence by opting to change the abstract to more clearly define the focus on socionatural hazards. We will update the abstract in the way foreseen in comment 3.07. |

| 3.06 | **Reviewer comment** |
|---|---|
| | In the Abstract section the sentences: "The interaction of social-ecological structures with these perceptions was analyzed, as well as the willingness to act, both individually and collectively, to address the challenges." and "This understanding is based on perceptions of social-ecological hazards and the complexity of perceived responsibilities and willingness to contribute to managing risks." provide overlapping information. I recommend to streamline or combine these sentences to avoid redundancy and improve clarity. |
| | **Author response** |
| | Good point. We agree and will leave the first sentence as is and delete the second one. The sentence after the deleted one (line 27/28) will then start with "This expanded understanding of lake level loss as a socionatural hazard therefore allows our analysis to shed more light on its practical implications…". |

| 3.07 | **Reviewer comment** |
|---|---|
| | The Abstract should mention all four research questions: (1) perceptions of social-ecological change, (2) the social structures that interact with these perceptions, (3) willingness to act and perceptions of responsibility, and (4) local practices for dealing with the challenges. However, only the first three appear to be addressed in the Abstract. It is recommended to |

include the fourth research question to ensure completeness and alignment with the study's stated aims.

**Author response**

Thank you for this suggestion. We will update the abstract to include the four research questions as follows, also considering the other reviewer comments:

"Groß Glienicker Lake and Sacrower Lake are two lakes in the Berlin-Brandenburg region that are facing significant challenges due to declining water levels associated with climate change. A mixed-method approach was employed, incorporating ethnographic research methods, a household survey and stakeholder workshops, in order to elicit perceptions on socionatural changes and challenges and address research questions pertaining to (1) perceptions of socionatural change, (2) social structures that interact with these perceptions, (3) willingness to act and perceptions of responsibility, and (4) local practices for dealing with the challenges. The analysis reveals that the hazard of lake level loss offers a prism through which diffracted socionatural challenges become visible, thus facilitating an understanding of social processes that shape the definition of the hazard beyond ecological aspects and the path forward in governing risks adaptively. This expanded understanding of lake level loss as a socionatural hazard therefore allows our analysis to shed more light on its practical implications in that focusing on purely technical solutions to maintaining or raising the water levels fails to orchestrate solutions to the social-ecological hazards."

| 3.08 | **Reviewer comment** |
|---|---|
| | Page 2, lines 43 and 49: I recommended to provide references to support these claims, such as newspapers articles, or press releases, or other relevant sources. |
| | **Author response** |
| | Thanks for the comment. There are different ways of describing ethnographically, since you interact with people on a daily basis, you refer to their actions and descriptions of past events. We included a reference to the website of the citizen initiative for the paragraph starting line 43. We will also include references to newspaper articles where possible. |

| 3.09 | **Reviewer comment** |
|---|---|
| | Page 4, line 102: I recommended to replace 'shared' with 'public,' 'collective,' or 'communal' to enhance clarity and better reflect the social dimensions of the lake space. |
| | **Author response** |
| | This will be replaced matching the new structure of this paragraph. |

| 3.10 | **Reviewer comment** |
|---|---|
| | Page 4, line 103: I recommended to specify what the self-efficacy refers to in this context to provide a clearer understanding of what individuals are expected to develop self-efficacy in. |

| | **Author response** |
|---|---|
| | It is very good that this has been brought to our attention. This sentence is indeed misleading. Due to local initiative, which is met with uncertainty about one's own influence on other people, self-efficacy must indeed be taken into account in governance structures. We will change it as follows:

Line 103: "Our analysis reveals a dynamic and changing approach to the lake as a shared and private space, the need for a governance that supports and enables residents' self-efficacy, and the problems of adopting one-size-fits-all strategies of action and communication." |

| 3.11 | **Reviewer comment** |
|---|---|
| | I recommended to use different font styles or formatting in Figure 1 to clearly differentiate between the federal state, district, and city names, in order to improve the readability of the figure. |
| | **Author response** |
| | Thanks for this suggestion. We will change figure 1 accordingly during the revision. |

| 3.12 | **Reviewer comment** |
|---|---|
| | In 2.2.1 Ethnographic methods section, I recommended to specify the number of interviews conducted for each category (neighbourhood representatives, household, and informal interviews) to provide a clearer breakdown of the data collection process. |
| | **Author response** |
| | Thank you for this comment. Informal interviews and neighbourhood representatives fall into the same category. We neither want to label conversations along certain types of categories nor show who represents what. Our aim is to give a general view on the discourse happening at the moment. Anonymity of research partners is a priority. |

| 3.13 | **Reviewer comment** |
|---|---|
| | For 2.2.3 Stakeholder workshops section, I recommended to specify whether the 8 to 10 representatives from the citizens' initiatives and public authorities participated in the four stakeholder workshops repeatedly or only once. Additionally, I recommend to identify the specific public authority institutions involved and clarify their roles in water management. This will provide a clearer understanding of the stakeholders' engagement and their respective contributions. |
| | **Author response** |
| | We agree and will specify in the text that representatives from the citizens' initiatives and public authorities participated in the four stakeholder workshops repeatedly and not only once. These were not necessarily the same persons at all four times. We will therefore add a table with the list of 11 organizations present throughout the workshop series and the |

sectors they represent (some individuals represented more than one organisation, which is why there are more organizations listed than individuals present at each workshop). Since only two of the 11 organizations were public authorities, however, we refrain from additionally specifying their specific roles in water management. This is what the table will look like (to be added after line 179):

**Table 1. Organisations represented throughout the workshop series**

| No. | Organisation | Sector |
|---|---|---|
| 1. | City Administration of Potsdam (Urban Development, Construction, Economy and Environment Division) | Public administration |
| 2. | Groß Glienicker Forum | Local political party |
| 3. | Local Advisory Council Groß Glienicke | Municipal council |
| 4. | Pro-Groß-Glienicker-See citizens' initiative | Civil society |
| 5 | Potsdam Institute of Inland Fisheries | Science |
| 6. | Sustainability Platform Brandenburg | Public sustainability network |
| 7. | Citizens' Advisory Council Sacrow | Civil society |
| 8. | Freies Ufer | Civil society |
| 9. | Potsdam Institute for Climate Impact Research | Science |
| 10. | Kladower Forum | Civil society |
| 11. | State Forest Enterprise Brandenburg (Forest enterprise Finkenkrug, Forest district Krampnitz) | Public administration |

| 3.14 | **Reviewer comment** |
|---|---|
| | I recommended to add the word 'future' in the title of Figure 3 for clarity, to indicate that the perceived challenges or dangers refer to future risks or concerns. |
| | **Author response** |
| | Thank you for the suggestion on precision. We will update the text in line 209 as follows: "This general awareness of current changes actually translates into concerns about the future for the Groß Glienicker lake, in which almost 90 % of the survey respondents reported the water level to be a large or very large future challenge for the lake (Fig. 3)"

 In addition, we will update the figure caption for Figure 3 to read (line 220): "Figure 3. Perceived future challenges or dangers for Groß Glienicker Lake (N = 644)." |

| 3.15 | **Reviewer comment** |
|---|---|
| | Page 9, line 230: Please expand on the claim that the resident's scientific community is divided on climate change. Specifically, clarify what this division refers to. |

| | |
|---|---|
| | **Author response** |
| | In this section, we provide the results of the survey concerning the respondents' perceptions on climate change skepsis. One item pertains to whether or not respondents believe that the scientific community is divided on climate change and not that the resident's scientific community is actually divided on climate change. Line 230 states this with "Moreover, one-third or less believe that the media exaggerate the effects of climate change and that the scientific community is divided on climate change." Therefore, we do propose any changes to the text beyond referring to the figure as in the following: "Moreover, as can be seen in Fig. 4, one-third or less believe that the media exaggerate the effects of climate change and that the 230 scientific community is divided on climate change." |

| | | |
|---|---|---|
| 3.16 | **Reviewer comment** | |
| | Regarding the question in Figure 4, the phrasing 'There are many different scientific opinions on climate change' is not accurate. The diversity of opinions is not relevant, as the discussion should focus on the scientific findings related to climate change, which are grounded in evidence-based analysis rather than opinions. | |
| | **Author response** | |
| | The questions in Figure 4 represent the three-question construct to elicit climate change skepticism among the respondents. Therefore, we are eliciting the perceptions of respondents in terms of to what degree they may be skeptical of climate change. We do not propose that there should or should not be a diversity of opinions on climate, and we do not discredit scientific findings related to climate change. For further clarification, if a respondent were not to be skeptical of climate change, they would respond with the answer "Very inaccurate". However, as can be seen in Figure 4, many respondents, despite evidence-based analysis, believe that there are different scientific opinions on climate change (i.e. approximately one-third answered slightly accurate, accurate or very accurate). | |

| | | |
|---|---|---|
| 3.17 | **Reviewer comment** | |
| | I recommended to revise the title of Figure 5 to include the word 'future' for consistency and clarity. | |
| | **Author response** | |
| | Thank you for this suggestion. We will update the figure caption of Figure 5 to read (line 250): "Figure 5. Average stakeholder perception of future challenges becoming smaller (-2), staying the same (0) or becoming larger (2) (N = 8)." | |

| | | |
|---|---|---|
| 3.18 | **Reviewer comment** | |
| | Page 12, lines 291-298: Why are the findings on improved water quality not mentioned as part of the preferences for future changes? | |

| | **Author response** |
|---|---|
| | Thank you for this question. We will add a sentence at line 293 to mention these results and update the sentences starting at line 292 as follows: "Nearly 100 % of respondents indicated the importance of water level stabilization as important or very important, and almost 80% of respondents perceived future improvements in water quality to be important or very important. Improved waste disposal was similarly ranked by over 90 % of respondents." |

| 3.19 | **Reviewer comment** |
|---|---|
| | Page 12, lines 279 to 287: the terms 'preference for future changes' and 'essential functions' appear to be used interchangeably. However, these concepts differ both conceptually and temporally—'preference for future changes' refers to desired or anticipated conditions, while 'essential functions' relate to current attributes or roles of the lake. It is recommended to clarify the distinction between these terms and ensure consistent usage throughout the text to avoid confusion. |
| | **Author response** |
| | Thank you for this comment. This has already been clarified in footnote number 2 of the original manuscript. |

| 3.20 | **Reviewer comment** |
|---|---|
| | The sections 3.2.2, 3.2.3, and 3.2.4 provide relevant information on the study site. However, the sources of this information are often not provided. I recommended to include appropriate references or clarify whether this information was obtained through primary data collection or secondary sources. This would strengthen the credibility and transparency of the study. |
| | **Author response** |
| | Thank you for this comment. We will add references where appropriate to clarify which information was obtained through primary data collection or secondary sources. |

| 3.21 | **Reviewer comment** |
|---|---|
| | Page 13, lines 310 to 319: include data on the average population age, income, and household size for both Berlin and Potsdam, along with appropriate references to support the claims that the studied population sample is older, wealthier, and has a larger household size. The demographic information will strengthen the validity of these comparisons. |
| | **Author response** |
| | Thank you for the suggestion to add further demographic information. We will update the text starting at line 311 as follows: "With regard to the quantitative survey, the sample from the study area was older (55 years on average) than the average population age of Berlin |

(42.8 years; Amt für Statistik Berlin-Brandenburg, 2024a) and Potsdam (43.2 years; Amt für Statistik Berlin-Brandenburg, 2024b). Overall, 66 % of the sample had at least a first degree and the average net income per month was over € 5,000, although the average monthly gross incomes of fully unemployed residents are approximately € 4,500 for Berlin and € 3,600 for Brandenburg (Amt für Statistik Berlin-Brandenburg, 2024c). Almost 60 % of the sample were employed at least part-time during the survey. The average household size was about 2.7 persons, higher than in Berlin (1.87 persons) and Potsdam (2,00 persons) (Statistisches Bundesamt, 2024). "

| 3.22 | **Reviewer comment** |
| --- | --- |
| | Table 1 does not provide a clear or understandable description in the first column. I recommend improving the table description to enhance clarity. Additionally, the meanings of AIC and BIC should be clarified, and the symbols *, **, and *** need to be explained. |
| | **Author response** |
| | We appreciate the suggestion to improve the table description. We will add the definitions below the table as caption for further information as follows: |
| | "Significance levels: ***p < 0.01, **p < 0.05, *p < 0.1 |
| | AIC: Akaike information criterion |
| | BIC: Bayesian information criterion" |
| | For improved clarity of the first column, we will add a column header with "(Interaction) variables of the model". Before the row starting with "Log likelihood", we will add a new header labeled "Model characteristics". |

| 3.23 | **Reviewer comment** |
| --- | --- |
| | Page 21, line 522: what does CI stands for? |
| | **Author response** |
| | It stands for "citizen's initiative". However, we will remove this abbreviation and write out "citizen initiatives" throughout the manuscript. |

| 3.24 | **Reviewer comment** |
| --- | --- |
| | In the Discussion section, I recommend to provide information on the uncertainty and limitations of the collected data. This could include potential sources of bias, sampling limitations, and any factors that may affect the reliability or generalizability of the results. |
| | **Author response** |
| | We appreciate the suggestion to describe the limitations. We will add a final paragraph to the discussion section starting at line 625 to briefly highlight the limitations as follows: |

"Our mixed-methods approach provided a nuanced understanding through different perspectives, but a few limitations do need to be mentioned. Despite the three different research approaches, it is possible that we were not able to garner all perceptions. The survey administration did not follow a systematic and random sampling approach, making the extrapolation of sample results to the population level difficult. In addition, online surveys inherently embed a potential sampling bias, although the results of survey were evaluated in collaboration with insights from the ethnographic work. Furthermore, although the stakeholder workshops included many representatives, it is possible that only interested parties participated in the workshop. Regarding both the ethnographic research and the stakeholder workshops, the interviewer bias could have led to potential issues; however, we managed this through joint interviews in initial stages of the research. Also, the potential influence of subjectivity on the results was further managed through transparency, mutual participation to different degrees by the researchers in the different approaches and active and iterative communication and collaboration in the elicitation of findings."

| 3.25 | **Reviewer comment** |
|---|---|
| | In the Conclusion section, I recommend to highlight the most important insights that directly address the four research questions. This will provide a clear and concise summary of the study's findings, ensuring that the key takeaways are easily accessible to readers. |
| | **Author response** |
| | Thank you for this valuable comment. We will make appropriate adjustments to the conclusion to ensure that the main insights in addressing the four research questions are highlighted. |

| 3.26 | **Reviewer comment** |
|---|---|
| | I recommend to add a description of the survey questions and the questions asked during the workshops to the Supplementary materials. This will allow readers to understand the data collection methods and the specific topics addressed in the study. |
| | **Author response** |
| | Given that the survey and the stakeholder workshops were carried out in German, we will add a footnote in the text at line 156 for the survey and line 170 for the stakeholder workshops with the following texts, respectively: "The survey can be made available upon request", "The structure and the guiding questions of the stakeholder workshop series can be made available upon request" (it is not possible to disclose all the questions asked during the workshops). |

| 3.27 | **Reviewer comment** |
|---|---|
| | Page 6, line 149: "...codes were grouped into thematic fields..." to "with codes grouped into thematic fields" |

| | **Author response** |
|---|---|
| | Thank you. We will change this in line 149. |

| | | |
|---|---|---|
| 3.28 | **Reviewer comment** | |
| | Page 20, line 480: "...they wanted to create awareness that you can't just go straight to the lake' (stakeholder representative)." to "..."you can't just go straight to the lake." (stakeholder representative)". | |
| | **Author response** | |
| | Thank you for indicating this error. We will make the correction as follows (line 480): "Rather they wanted to create awareness that "you can't just go straight to the lake" (stakeholder representative). | |

| | | |
|---|---|---|
| 3.29 | **Reviewer comment** | |
| | Page 20, line 482: "...the aum is not to prohibit..." What is aum? | |
| | **Author response** | |
| | Thank you for indicating this error. This will be corrected as follows (line 479):"He explained that the aim is not to prohibit potential swimmers from coming to the lake, and hence they leave gaps between wooden fences." | |

| | | |
|---|---|---|
| 3.30 | **Reviewer comment** | |
| | Page 23, line 566: "...responsibilities should be more clearly distributed between citizens, administration and politics." To "...between citizens, administrative bodies, and political decision-makers." | |
| | **Author response** | |
| | Thank you for indicating this error. This will be corrected as follows (line 565): "These local practices of concerned citizens on the ground indicate that concerted efforts on the part of the residents are appreciated, but also that responsibilities should be more clearly distributed between citizens, administrative bodies, and political decision-makers." | |